# Chiral expression from molecular to macroscopic level via pH modulation in terbium coordination polymers

Jian Huang[1], Hong-ming Ding[2,3], Yan Xu[1], Dai Zeng[1], Hui Zhu[1], Dong-Mian Zang[1], Song-Song Bao[1], Yu-qiang Ma[2,3] & Li-Min Zheng [1]

Chiral expression from the molecular to macroscopic level is common in biological systems, but is difficult to realise for coordination polymers (CPs). The assembly of homochiral CPs in both crystalline and helical forms can provide a bridge for understanding the relationship between the molecular and macroscopic scales of chirality. Herein, we report homochiral helices of [Tb($R$- or $S$-pempH)$_3$]•2H$_2$O (**R-** or **S-1**) (pempH$_2$ = (1-phenylethylamino) methylphosphonic acid) and their crystalline counterparts (**R-** or **S-3**), which are formed at different pH of the reaction mixtures under hydrothermal conditions. By combining the experiments and molecular simulations, we propose that the formation of helices of **R-1** or **S-1** occurs via a hierarchical self-assembly route, which involves twisted packing due to the geometric incompatibility of the different types of chains. The observed chiral transcription from molecules to morphologies is significant for understanding bio-related self-assembly processes on the nano- to macro-scale.

[1] State Key Laboratory of Coordination Chemistry, School of Chemistry and Chemical Engineering, Collaborative Innovation Center of Advanced Microstructures, Nanjing University, Nanjing 210023, China. [2] National Laboratory of Solid State Microstructures and Department of Physics, Collaborative Innovation Center of Advanced Microstructures, Nanjing University, Nanjing 210093, China. [3] Center for Soft Condensed Matter Physics and Interdisciplinary Research, Soochow University, Suzhou 215006, China. Jian Huang, Hong-ming Ding and Yan Xu contributed equally to this work. Correspondence and requests for materials should be addressed to S.-S.B. (email: baososo@nju.edu.cn) or to Y.-q.M. (email: myqiang@nju.edu.cn) or to L.-M.Z. (email: lmzheng@nju.edu.cn)

In biological systems, chirality is widely expressed from small molecules such as L-amino acids and D-sugars to macroscopic helices such as double helical DNA and α-helical proteins. The helical assembly of large biomolecules not only enables their biological functions required to sustain life, but also allows for the development of novel applications such as bio-computing, bio-catalysis, biomedicine and material science[1–8]. On the other hand, DNA and protein crystallisation is increasingly demanding in order to elucidate the structure–function relationships at the atomic and molecular scale in the biological systems. Despite the tremendous difficulties in assembling the helical bio-motifs into three-dimensional crystalline lattices, significant progress has been witnessed in obtaining diffraction-quality protein, peptide and DNA crystals and in establishing their roles as catalytic materials, drug delivery vehicles and chemical reaction vessels[9–13].

DNA and protein crystallisation requires the enhancement of interactions between helical motifs through non-covalent or covalent bondings. One of the most promising routes is metal-mediated head-to-tail assembly by taking the advantage of the metal–ligand interactions at the ends of the growing crystal[14]. In this case, the resulted crystals can be viewed as 'biocoordination polymers' in which the helical biomacromolecules are connected by metal ions into polymers with ordered structures.

In non-biological systems, coordination polymers (CPs) refer to a broad range of metal–organic compounds with one-, two- and three-dimensional structures, composed of metal ions as nodes and small organic ligands as linkages. These materials usually possess precise structures that can be designed and constructed through careful selection of metal ions and ligands[15,16]. As a subclass of CPs, homochiral CPs are very attractive in the fields of enantioselective catalysis and separation, chiral sensing, nonlinear optics and multifunctional materials[17–20]. In contrast to the biomacromolecules that are difficult to crystallise, most homochiral CPs appear as crystalline materials with well-defined structures, and the chirality is typically expressed only at the molecular level. Examples of homochiral CPs with helical morphologies are extremely rare[21–26], despite the great efforts devoted to the fabrication of artificial chiral nanoarchitectures in order to mimic the natural biological systems[27–33]. By utilising chiral amino acids, Tang et al. obtained homochiral Ag(I)/cysteine helical nanobelts in which the chirality transcription occurs from cysteine molecule to the assembly entities[21]. The proposed mechanism involved the merging of nonhelical nano-belts of Ag(I)/cysteine layers through lateral attachment, which developed into hierarchical helices with a specific twist direction. Huang et al. isolated right-handed Ca-cholate helical nanori-bbons, which were further used as templates for the fabrication of helical inorganic nanomaterials[22]. The formation mechanism was supposed to involve the twist of supramolecular layers composed of cholate bilayer strips connected via calcium–carboxyl coordination. In all these cases, crystalline forms of the same materials were not obtained. Therefore, structural illustration of these helices was not very clear due to the absence of single crystal data, although theoretical calculations were conducted to propose the structures. The scarce examples demonstrate the challenges remaining in the construction of homochiral helical CPs. More challenging is the assembly of homochiral CPs in both crystalline and helical forms, which can provide a bridge for understanding the relationship between the molecular and macroscopic scales of chirality.

In nature, the macroscopic scale biomolecules contain supra-molecular nanostructures, hierarchically organised via weak interactions such as π–π stacking, hydrogen bonding and hydrophobic interactions. Hence, the selection of suitable organic ligands and the control of weak interactions between the chain or layer motifs could be a key for the construction of homochiral CPs in crystalline and/or helical morphologies.

The optically active amino phosphonic acids are known to play important roles in biological activities[34,35]. They can dissociate in aqueous solutions with the release of one or two protons, depending on the pH of the media. Each phosphonate group can bind up to nine metal atoms with versatile coordination modes[36,37]. Modification of the organic groups can provide additional sites for metal coordination, hydrogen bonding and/or π–π stacking. Previously, we found that enantiopure R- or S-(1-phenylethylamino)methylphosphonic acid (R- or S-pempH$_2$) can react with metal ions to form layered[38,39] or nanotubular[40,41] structures. Considering that trivalent lanthanide ions show high kinetic lability and a lack of stereochemical preference, herein we report on the reactions of Tb$^{3+}$ nitrate and R- or S-pempH$_2$ under hydrothermal conditions. Interestingly, homochiral coordination polymers of [Tb(R- or S-pempH)$_3$]•2H$_2$O with both helical (R-, S-1) and crystalline (R-, S-3) morphologies are obtained by pH modulation of the reaction mixture. Results based on both the experiments and molecular simulations reveal that the formation of helices of R-1 or S-1 occurs via a hierarchical self-assembly route, which involves twisted packing due to the geometric incompatibility of the different types of chains.

## Results

**Synthesis and characterisation of homochiral helices.** An aqueous solution of Tb(NO$_3$)$_3$ and R-pempH$_2$ (Fig. 1) in molar ratio 1:5 was adjusted to pH 3.1 using 0.5 M NaOH and heated hydrothermally at 120 °C for 2 days. After cooling to room temperature, both the flocculent precipitates on the top and the powder precipitates on the bottom of the vessel were obtained. The flocculent precipitates were collected manually, and the other precipitates were collected by suction filtration. Powder X-ray diffraction (PXRD) measurements confirmed that the two pre-cipitates were the same material, which is referred to hereafter as R-1. When S-pempH$_2$ was used as the starting material, the final product, referred to as S-1, showed an identical PXRD pattern to that of R-1 (Supplementary Fig. 1).

Figure 2 shows the scanning electron microscope (SEM) images of the as-synthesised flocculent precipitates of R-1 and S-1. R-1 exhibited a pure right-handed helical morphology, whereas S−1 showed a pure left-handed helical morphology. The diameters of the helices were ca. 6–9 μm and the lengths ranged from 100 to 300 μm. The pitches of the helices were ~20–30 μm, and the pitch angles ranged from 70 to 80° for most helices (Supplementary Fig. 2).

The mirror image relationship of R-1 and S-1 was reflected not only in their morphologies, but also in their optical properties. The infrared spectra (IR) of both R-1 and S-1 revealed sharp peaks in the 900–1200 cm$^{-1}$ region, which were attributed to the stretching vibrations of the −PO$_3$ group (Supplementary Fig. 3). Solid-state circular dichroism (CD) spectra revealed Cotton effects centred at 262 nm, originating from the ligand; opposing symmetries were observed for R-1 and S-1, which indicated that the chirality was transferred from the ligand to the inner structure of the helices (Fig. 2e and Supplementary Figs. 4–6). The chemical

**Fig. 1** Molecular structures of ligands. Molecular structures of (R)-pempH$_2$ (**a**) and (S)-pempH$_2$ (**b**)

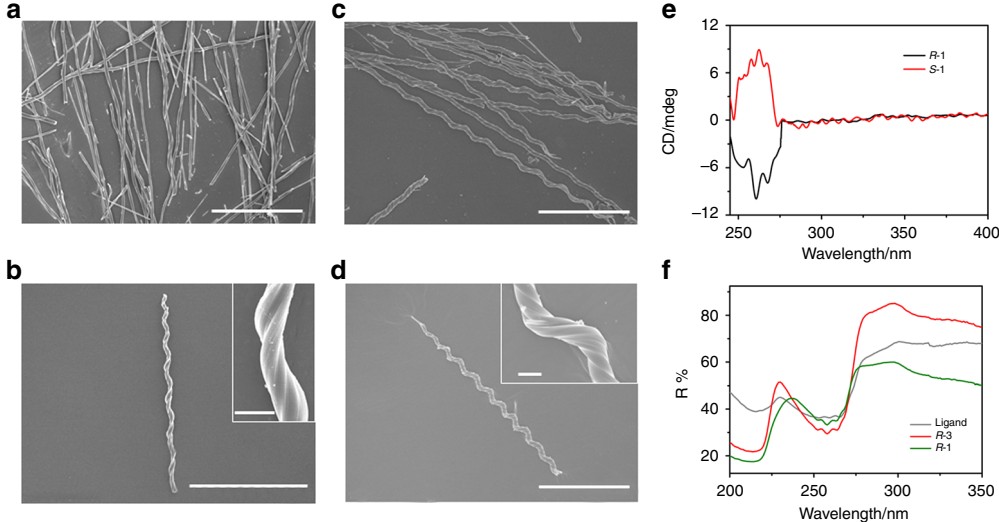

**Fig. 2** SEM images, UV–Vis and CD spectra of helices. **a**, **b** SEM images of *R*-1. **c**, **d** SEM images of *S*-1. Scale bar: 100 μm, inset: 5 μm. **e** CD spectra of *R*-1 and *S*-1. **f** Reflectance UV–Vis spectra of *R*-pempH₂, helices of *R*-1 and rod-like crystals of *R*-3

composition, as determined by the energy dispersive X-ray spectroscopy (EDX) (Supplementary Figs. 7 and 8 and Supplementary Table 1), revealed a Tb/P atomic ratio of ~1:3 in both *R*-1 and *S*-1. Based on elemental (Supplementary Table 2) and EDX analyses, the molecular formulae of the homochiral helices were proposed to be h-Tb(*R*-pempH)₃•2H₂O for *R*-1 and h-Tb(*S*-pempH)₃•2H₂O for *S*-1 (h means helical). The number of water molecules was confirmed by thermal analyses (Supplementary Fig. 9).

**Influence of reaction conditions on the helix formation.** To determine the effects of the reaction conditions on the formation of the helices, a systematic study was performed on the Tb/*R*-pempH₂ system by varying the pH, reaction time, temperature, as well as the cations and anions in the reaction mixture.

First, Tb(NO₃)₃ was allowed to react with *R*-pempH₂ under hydrothermal conditions (120 °C, 2 d) at different pH values, which were appropriately adjusted using 0.5 M NaOH (Supplementary Fig. 10). A transparent solution without any precipitates was obtained when the pH was sufficiently low (1.5–2.4). At pH 2.5–2.7, block-like colourless crystals of (H₃O) [Tb₃(*R*-pempH₂)₂(*R*-pempH)₇][Tb₃(*R*-pempH₂)(*R*-pempH)₈] (NO₃)₄•11H₂O (referred to hereafter as *R*-2) formed (Supplementary Fig. 11), together with a small amount of white precipitate of *R*-1. Higher pH value (2.7–2.9) resulted in a mixture of block-like crystals of *R*-2 and helices of *R*-1. When the pH was 3.0–3.2, only helices of *R*-1 were obtained. When the pH was further increased to 3.3–3.6, both helices of *R*-1 and rod-like crystals of c-Tb(*R*-pempH)₃•2H₂O (referred to hereafter as *R*-3, c means crystalline) were isolated. Finally, at pH 3.7–4.5, only *R*-3 was isolated. These results clearly indicated that the pH of the reaction mixture was essential for the formation of the helices. To investigate whether helices of *R*-1 could be crystallised, we added additional 0.5 M NaOH solution to the helices in mother liquid (obtained at pH = 3.1) until the final pH was about 4.5, and put the autoclave back into the oven at 120 °C. After 2 days, the white powdered helices were transformed into the rod-like crystals of *R*-3. Therefore, we propose that the structure of the helices of *R*-1 is closely related to the structures of *R*-2 and *R*-3.

Next, hydrothermal reactions of Tb(NO₃)₃ and *R*-pempH₂ (pH 3.1) were conducted at 120 °C for different periods of time (1–24 h). As shown in Supplementary Figs. 12 and 13, helices of *R*-1 were present even at 1 h. When the reaction time was increased,

both the length and the diameter of the helices increased. When the reaction time was sufficiently long, e.g., after 8 h, the morphology of the final products did not change significantly.

We next asked whether the presence of metal ions other than Na⁺ could affect the final products. To test this, we conducted the hydrothermal reactions of Tb(NO₃)₃ and *R*-pempH₂ (pH = 3.1) at 120 °C for 2 d, and used other alkali or alkaline earth metal hydroxides, such as KOH and Ca(OH)₂, to adjust the pH (instead of NaOH) or added Ba(NO₃)₂/Sr(NO₃)₂ to the reaction mixture. The SEM images and the PXRD patterns of the final products showed that generally the cations had little influence on the formation of the helices (Supplementary Figs. 14 and 15). However, the addition of a particular cation did affect the morphology of the helices. Supplementary Fig. 14 clearly shows that helices formed in the presence of Ba²⁺ had smaller helical pitches.

The effect of anions was also studied. Different terbium salts, such as TbCl₃, Tb(OAc)₃ and Tb(ClO₄)₃, were used as representative to react with *R*-pempH₂ (pH = 3.1, adjusted using 0.5 M NaOH) at 120 °C for 2 d. Surprisingly, no helices were found in the final products, as shown in Supplementary Fig. 16. In other words, among the tested anions, only NO₃⁻ could induce the formation of helices.

Finally, hydrothermal reactions of Tb(NO₃)₃ and *R*-pempH₂ (pH 3.1) were performed for 20 h but at different temperatures (80–160 °C). Supplementary Fig. 17 showed that helices of *R*-1 were obtained when the reaction temperature reached 100–140 °C, but the pitches of the helices were significantly different.

Apparently, helices of *R*-1 can form in a suitable temperature range. The reaction time and addition of alkali or alkaline earth metal ions could affect the size and morphology of the helices. The helix formation is pH-sensitive and anion-dependent. Notably, helices formed at ca. pH 3.1, whereas block- or rod-like crystals of *R*-2 and *R*-3 were isolated below or above this pH. An accurate determination of the crystal structures can help reveal information about the formation mechanism of the helices. We next examined the crystal structures of the compounds we obtained.

**Structural description of homochiral block-like crystals of *R*-2.** Figure 3 shows the structure of the freshly prepared *R*-2 (Supplementary Figs. 18 and 19), which crystallises in the monoclinic system, chiral space group *P*2₁ (no. 4) (Supplementary Table 3).

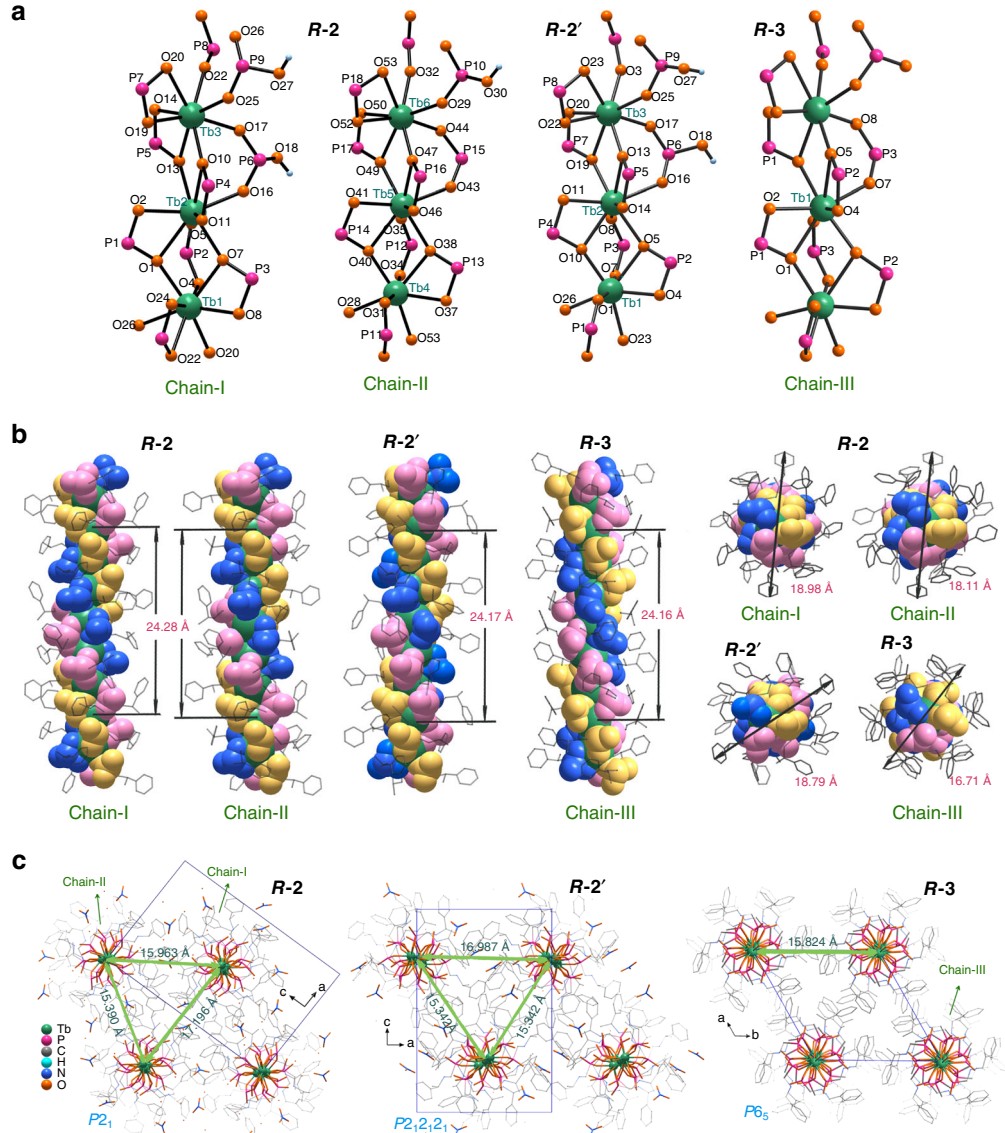

**Fig. 3** Crystal structures of **R-2**, **R-2'** and **R-3**. **a** Chain structures with atomic labelling in **R-2**, **R-2'** and **R-3** along the crystallographic $2_1$ or $6_5$ screw axis. The N and C atoms are omitted for clarity. **b** Side and top views of the chains in structures **R-2**, **R-2'** and **R-3**, which contain left-handed triple strands. The pitches and diameters of each helical chain are also given. **c** Packing diagrams of structures **R-2**, **R-2'** and **R-3** viewed along the crystallographic $b$- or $c$-axis. The interval of adjacent chains are defined by the distances between the relative $2_1$ axis along the $b$-direction in **R-2** and **R-2'** or $6_5$ axis along the $c$-direction in **R-3**

Two types of chains are observed in the structure (Fig. 3a). Chain-I has the composition $[Tb_3(R\text{-pempH}_2)_2(R\text{-pempH})_7]^{2+}$ and contains three distinct Tb atoms (Tb1, Tb2 and Tb3). Each Tb is eight coordinated by oxygen atoms from six phosphonate ligands [Tb–O: 2.306(10)–2.650(11) Å, O–Tb–O: 58.0(4)–169.3(3)°] (Supplementary Fig. 20 and Supplementary Table 4). Neighbouring $Tb^{3+}$ ions are triply bridged by two $\mu_3$-O(P) and one O–P–O units, forming infinite chains that run along the $b$-axis. The Tb···Tb distances are 4.025(1) Å for Tb1···Tb2, 4.139(1) Å for Tb2···Tb3, and 4.061(1) Å for Tb1···Tb3$^i$ (i, 1−x, −0.5+y, 1−z). The Tb···Tb···Tb angles are in the range of 171.8(1)–175.3(1)°. The chain contains triple helical strands of −Tb–O–P–O− with a pitch of 24.28 Å and a diameter of 18.98 Å (Fig. 3b). There are two different $R$-pempH$_2$ and seven $R$-pempH$^-$ in chain-I; the two $R$-pempH$_2$ zwitterions are each singly protonated at O18 and O27, which are hydrogen bonded to two $NO_3^-$ anions [O18···O57: 2.57(2) Å; O27···O63: 2.607(18) Å] (Fig. 4).

Chain-II has the composition $[Tb_3(R\text{-pempH}_2)(R\text{-pempH})_8]^+$. Unlike chain-I, only one $R$-pempH$_2$ is found in chain-II, which is singly protonated at O30 (Fig. 3a). The three crystallographically distinct Tb atoms are either 7- (for Tb4) or 8-coordinated (for Tb5, Tb6). Within the chain, the Tb atoms are triply bridged by two $\mu_3$-O(P) and one O–P–O units to form a trimer. Between the trimers, the Tb atoms (Tb4 and Tb6) are connected by one $\mu_3$-O (P) and two O–P–O units. As a result, the Tb4···Tb6$^{ii}$ (ii, −x, −0.5 +y, 2−z) distance [4.390(1) Å] is much longer than those of Tb4···Tb5 [3.931(1) Å] and Tb5···Tb6 [3.911(1) Å]. The Tb5···Tb6···Tb4 angle [164.7(1)°] is much smaller than the other Tb···Tb···Tb angles [173.2(1)–174.4(1)°]. Chain-II is more twisted than chain-I. Triple helical strands of −Tb–O–P–O− are found in chain-II, with the same pitch as that in chain-I and a diameter of 18.11 Å (Fig. 3b).

Chain-I and chain-II are packed in the lattice with interchain distances of 15.390, 15.963 and 17.196 Å (Fig. 3c). Four

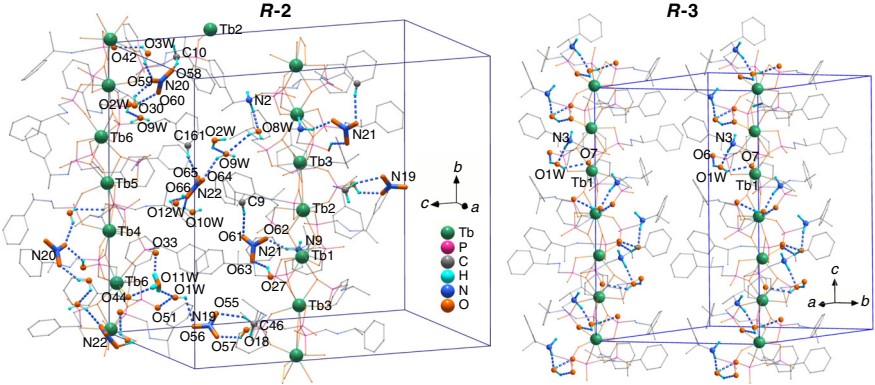

**Fig. 4** Hydrogen bonds in **R-2** and **R-3**. Hydrogen bonds for $NO_3^-$ anions, $H_3O^+$ cations, and/or lattice water molecules are highlighted

crystallographically distinguised $NO_3^-$ anions (N19, N20, N21 and N22) and one $H_3O^+$ (O11W) exist as charge-balancing counterions between the chains. The $H_3O^+$ cations are hydrogen bonded to the phosphonate oxygen atoms from chain-II and lattice water molecules (Fig. 4). The $NO_3^-$ anions are involved in the hydrogen bond network with the protonated phosphonate oxygen (O18, O27 and O30), amino nitrogen, $-CH_2-$ and phenyl groups as well as lattice water molecules (Supplementary Table 5).

**Structural description of homochiral block-like crystals of R-2′.** Interestingly, compound **R-2** underwent a solid state structural transformation soon after exposure to air at room temperature, forming Tb₃(R-pempH₂)₂(R-pempH)₇(NO₃)₂•2H₂O (**R-2′**), which crystallises in space group $P2_12_12_1$ (no. 19) (Supplementary Figs. 21–24 and Supplementary Tables 3 and 6). The structural transformation involved the change of the proton position from $H_3O^+$ to the phosphonate group. Although the chemical composition of **R-2′** is similar to that of chain-I in **R-2**, its structure is closer to that of chain-II, as shown in Figures 3a and 3b. Hydrogen-bond interactions are found among the protonated phosphonate oxygen (O18 and O27), the $NO_3^-$ anions and the amino groups (Supplementary Fig. 25 and Supplementary Table 7), thus stabilising the 3D supramolecular network structure (Fig. 3c).

**Structural description of homochiral rod-like crystals of R-3.** Unlike **R-2** and **R-2′**, c-Tb(R-pempH)₃•2H₂O (**R-3**) displays a neutral chain structure, crystallising in space group $P6_5$ (no. 170) (Supplementary Figs. 26–28 and Supplementary Table 3). Only one Tb is crystallographically distinguished, which is 8-coordinated by oxygen atoms from six R-pempH⁻ [Tb–O: 2.28 (2)–2.632(16) Å, O–Tb–O: 58.1(5)-162.2(5)°] (Supplementary Fig. 29 and Supplementary Table 8). The equivalent Tb atoms are triply bridged by two $\mu_3$-O(P) and one O–P–O units into a helical chain running along the c-axis (named as chain-III) (Fig. 3a). The Tb⋯Tb separation is 4.043(1) Å. The Tb⋯Tb⋯Tb angle is 174.8 (1)°. Chain-III also contains left-handed triple helical strands (Fig. 3b). Remarkably, although the pitch of the helical chains in **R-3** is only slightly smaller than those in **R-2** (24.16 Å vs. 24.28 Å in **R-2**), there is a significant reduction in the chain diameter (16.71 Å vs. 18.11–18.98 Å in **R-2**). The adjacent helical chains are packed into a 3D supramolecular framework via hydrogen bond interactions (Supplementary Table 9) with an interchain distance of 15.824 Å (Fig. 3c).

**Comparison of the rod-like crystals of R-3 and helices of R-1.** The chemical composition of **R-3** is identical to that of **R-1**, suggesting that the two materials could be closely related to each other. Indeed, the IR spectra of **R-1** and **R-3** were identical, and

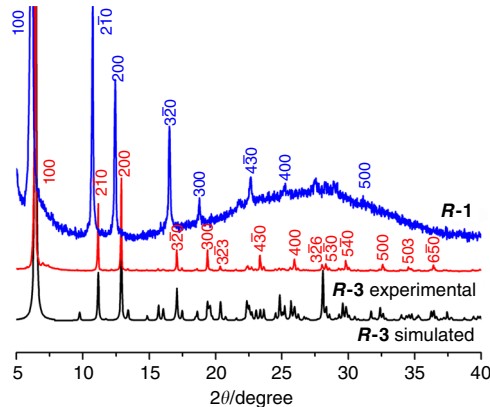

**Fig. 5** PXRD patterns of **R-1** helices and **R-3**. The peaks are marked with the corresponding Miller indices. Simulated PXRD pattern of compound **R-3** is also shown for comparison

the PXRD patterns of **R-1** and **R-3** were similar, as shown in Fig. 5. The reflections of (h00) for **R-1** appeared as a series of strong and equidistant peaks with h up to 5 due to preferential orientation, which indicated that the helices possessed long-range intermolecular order and had a similar structure as that of the rod-like crystals of **R-3**. The diffraction peaks can be indexed by using TOPAS 4.2 programme[42], giving a set of unit cell parameters with space group $P6_5$, $a = 16.28$ Å, $c = 24.24$ Å and $V = 5564.9$ Å³ for **R-1** (Supplementary Fig. 30). Compared with those in **R-3**, all peaks for **R-1** helices were shifted to lower angles and the cell volume was expanded (Supplementary Table 10). As such, a larger interchain distance can exist in the helices of **R-1** than in the crystals of **R-3**. The average distance between two layers assembled by the 1D helical chains in the bc plane was estimated to be 14.1 Å in the helices of **R-1**, much larger than that in the crystals of **R-3** (13.7 Å). This difference may originate from the distortion of the 1D chains during the self-assembly process. According to the experimental data, we conclude that the helices of **R-1** have a high degree of crystallinity, with almost the same structure at the molecular level as that of rod-like crystals of **R-3**, but at the macroscopic level different from **R-3**. To the best of our knowledge, this is the first example of coordination polymers that show both crystalline and helical morphologies.

**Characterisation of intermediate self-assembly states.** In order to get more insights into the formation mechanism of the chiral morphology, we monitored the solid products after hydrothermal reactions of Tb(NO₃)₂ and R-pempH₂ (pH = 3.1) at 120 °C for different period of time. The PXRD measurements revealed that

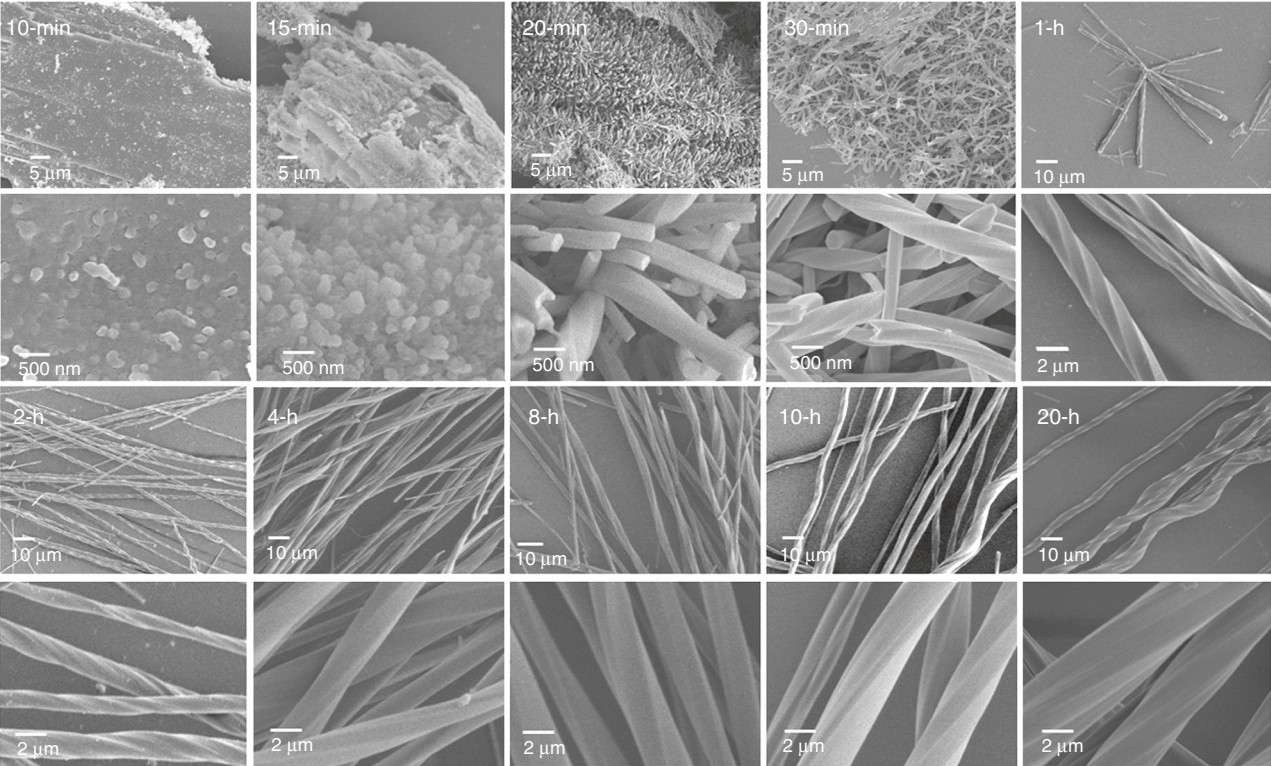

**Fig. 6** SEM images of the intermediate self-assembly states. SEM images of the solid products after hydrothermal reactions of Tb(NO$_3$)$_3$ and $R$-pempH$_2$ (pH ~ 3.1) at 120 °C for different period of time

the 0-min product (before hydrothermal reaction) contained only the ligand of $R$-pempH$_2$. After 10 min, helices of **R-1** appeared together with the ligand, evidenced by the emergence of a peak at a lower angle of 6.183°. The two phases, e.g., **R-1** and the solid $R$-pempH$_2$, coexisted in the reaction mixture even after 1 h. When the reaction time reached 2 h and above, pure phases of **R-1** can be observed (Supplementary Fig. 31).

The IR spectra can recognise the presence of un-coordinated nitrate anions by the peak at ca. 1385 cm$^{-1}$, which was significant in the case of pure **R-2**. As shown in Supplementary Fig. 32, the intensity of this peak was markedly increased when the reaction time reached 4–6 h, indicating that the NO$_3^-$ anion and hence the positively charged chains (chain-I, -II in **R-2**) could be involved in the helices of **R-1**. The increase of the peak intensity cannot be identified when the reaction time was less than 2 h or more than 8 h, possibly due to the interference of the ligand and/or the presence of neutral chains in the helices of **R-1**. The results suggested that the amount of positively charged chains in **R-1** could be very small compared with that of the neutral ones. This may explain the fact that both the PXRD pattern and the chemical composition of the helices of **R-1** are close to those of compound **R-3**.

The SEM images give a clear visualisation about the morphology of the solid products. The 0-min product contained sheets of $R$-pempH$_2$ covered by some amorphous nano-particles of size 100–300 nm (Supplementary Fig. 33). The nano-particles without helical morphologies were also observed in the 10-min product (Fig. 6), although PXRD result suggested the helix formation after 10 min of reaction. In the 15-min product, aggregates of both nano-particles and nanorods were found. Still, the nanorods did not show helical morphologies, but their Tb/P ratio was close to 1/3. For comparison, the Tb/P ratio in the nano-particle area was ca. 1/7 (Supplementary Fig. 34).

Interestingly, small helices appeared in the 20-min product like an actinian, the widths and lengths of which were ca. 150–500 nm and 2–5 µm, respectively. The Tb/P atomic ratio was ca. 1/3 for the helices, but ca. 0/3 for the bottom area without helices (Supplementary Fig. 35). After reacting for 30 min, aggregates of helices were observed like the bird's nest. Compared with the 20-min product, however, the lengths of the helices (ca. 2.05–11.78 µm) become longer, and the widths (ca. 0.19–0.97 µm) become wider (Supplementary Fig. 36). For the 1-h product, starfish-like aggregates of helices can be recognised together with the separate ones. Meanwhile, the lengths (ca. 6.37–119.45 µm) and the widths (ca. 0.19–2.26 µm) of the helices increased further, from nanometre to micrometre scale. After 2–8 h of hydrothermal reactions, the helices of **R-1** appeared separately, and the average length and width of the helices increased with increasing reaction time. Notably, the widths of the helices do not change significantly after 8 h of reactions, although the lengths of the helices increased continuously (Fig. 6 and Supplementary Figs. 37 and 38).

Noting that the growth of helices occurred on the surface of the ligand, the un-dissolved $R$-pempH$_2$ could play two roles during the helix formation: (1) it served as a buffer against the pH change in solution during the self-assembly process. The pH of a saturated solution of $R$-pempH$_2$ was ca. 3.5 (Supplementary Fig. 39). The coordination of $R$-pempH$_2$ with Tb$^{3+}$ would release the protons, and then decreased the pH in solution (Supplementary Fig. 40). Thus, the dissolve of solid $R$-pempH$_2$ could help maintaining the pH of solution in a suitable range (ca. 3.1) during helix formation. (2) It served as nucleation centres at which the helices of **R-1** grew.

From the above experimental results, we can conclude that the helices of **R-1** formed on the surface of the un-dissolved ligand of $R$-pempH$_2$, forming nano-particles first, then the nanorods, and

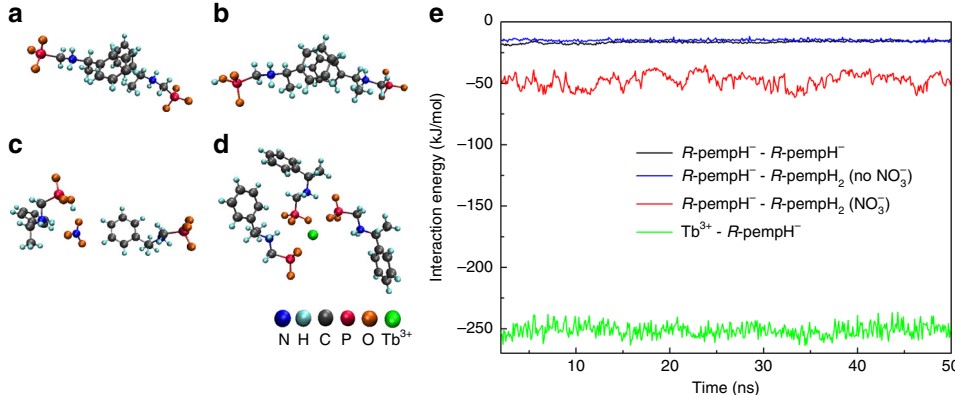

**Fig. 7** The setup and result of all-atom molecular dynamics simulations. The initial conformation for different systems (the coordinates are adopted from the experimental crystal data): **a** two $R$-pempH$^-$ molecules; **b** one $R$-pempH$^-$ molecule and one $R$-pempH$_2$ molecule; **c** one $R$-pempH$^-$ molecule and one $R$-pempH$_2$ molecule in the presence of NO$_3^-$; **d** one Tb$^{3+}$ and three $R$-pempH$^-$ molecules. The water molecules and counterions are not shown for clarity. **e** Interaction energy among different molecules in the above systems

finally the helices of **R-1**. The growth of the helices followed a hierarchical process with the length direction growing much faster than the width direction. Both the neutral chains of **R-3** and the positively charged chains of **R-2** were involved in the helix formation process.

**Molecular modelling of the formation mechanism of helices.** To better understand the underlying mechanism of the helix formation, we applied the all-atom molecular dynamics simulation to investigate the interaction energy among Tb$^{3+}$, $R$-pempH$^-$, $R$-pempH$_2$, and also used the coarse-grained Brownian dynamics (BD) simulation to study the self-assembly of chain-II and chain-III under different cases.

As shown in Fig. 7, the interaction energy between Tb$^{3+}$ and $R$-pempH$^-$ was much larger (about 15 times) than that between $R$-pempH$^-$ and $R$-pempH$^-$ (or $R$-pempH$_2$), indicating that the growth rate along the chain direction should be faster than that along the side direction. However, in the presence of NO$_3^-$, since there existed hydrogen bonds among NO$_3^-$, $R$-pempH$^-$ and $R$-pempH$_2$, the interaction energy increased a lot (about three times). As a result, the difference of the growth rate between chain direction and side direction would not become very obvious and the length and diameter of assembly would be comparable, which may help explain the block crystal in **R-2** system.

To better clarify the experimental observations, we further used Brownian dynamics simulation to investigate the system from the mesoscopic view (Fig. 8). In the case of pure chain-III, rod-like chains with one or several monomers in the diameter were observed (Fig. 9a). While in the case of pure chain-II, due to the strong side interaction, block-like aggregates with similar length and height were observed (Fig. 9b). These results were in accordance with the inference by all-atom simulation, and consisted with the experimental findings. More importantly, in the case of the mixture of chain-III and chain-II (the ratio is about 4:1), the curved or twisted chains were found (Fig. 9c). Since the molecular symmetries of chain-III ($P6_5$) and chain-II ($P2_1$) are different, when chain-II bound to chain-III along the axis, the assembled chain would become a bit curved instead of linear growth in **R-3** system. Moreover, since the side interaction between chain-II and chain-II was much larger than that between chain-III and chain-II/chain-III, chain-II preferred to aggregate with each other, which further made the growth along the chain direction more curved. Notably, the curved chain does not mean that it is chiral. But the curvature of the chain must be the

prerequisite for the helix formation; in other words, pure linear chains can never form helices. As shown in Supplementary Fig. 41, the correlation function of the linear chain was totally different from that of the curved chain and the helical chain. Recently Grason et al.[43] also demonstrated theoretically that chiral filaments may occur due to the frustration of inter-filament spacing when there exist many self-twisting curved chains or bundles. We should mention that due to the limitation of present computing technology, the length scale of the obtained chains in the simulations was about tens of (at most one hundred) nanometres, which was far from the length scale of the helices (tens of micrometres) in the experiments. Thus the chains here were still not the direct evidence of macroscopic helices. Nevertheless, the present simulation provides some hints or possible explanations for the reason why the macroscopic scales of chirality could be observed in this coordination polymer system.

Additionally, if there did not exist the NO$_3^-$ ions in the system, namely, the side interaction between chain-II and chain-II was the same as that between chain-III and chain-II/chain-III, the aggregation of chain-II in the assembly would not become obvious, and finally the assembly was nearly linear chain instead of twisted one (Fig. 9d), which indicated the importance of NO$_3^-$ in the chiral assembly and was in good agreement with the experimental findings.

Besides, we also found that if the mixture ratio of chain-III and chain-II was lower (e.g., 1:1), since the side interaction between chain-II and chain-II was large, block-like aggregates were again observed (Fig. 9e). On the contrary, if the mixture ratio of chain-III and chain-II was much greater (16:1), since there existed no enough chains-II to induce the curved growth of chains-III, the twist or curve of the assembly was not obvious (Fig. 9f), indicating that the ratio of chain-III and chain-II was also of great importance in helices formation. Considering that pH is related to the ratio of chain-III and chain-II in the system, our simulation results here could also clarify the role of pH in the experiments.

**Discussion**

We described the generation of homochiral helices of **R-1** or **S-1** under hydrothermal conditions when the pH of the reaction mixture was about 3.1. Below or above this pH, crystalline compounds **R-2** (or **S-2**) and **R-3** (or **S-3**) were isolated. Such a pH-responsive morphology change is reminiscent of biological systems, where a pH is often crucial for conformational changes

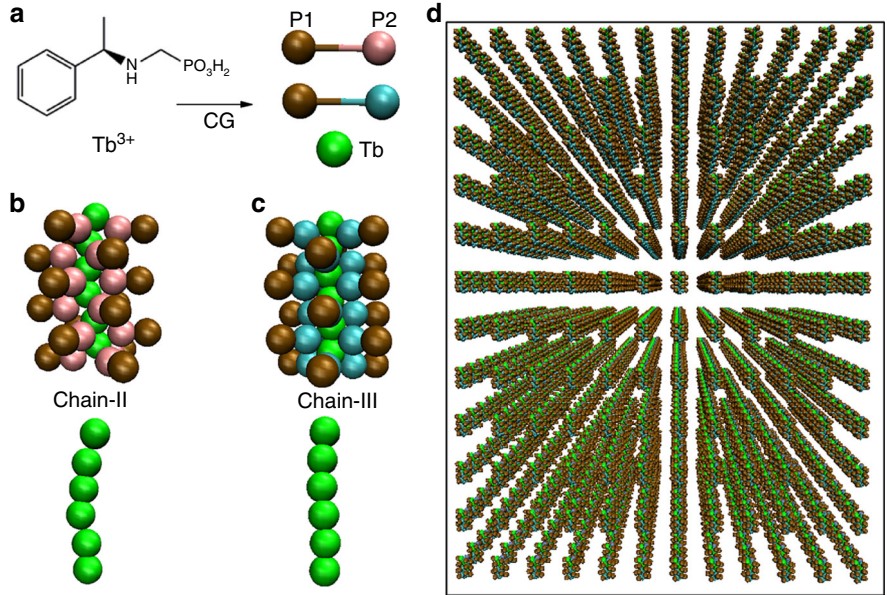

**Fig. 8** The setup of Brownian dynamics simulations. **a** Schematic illustration of the CG models for $R$-pempH$^-$ ($R$-pempH$_2$) molecule and Tb$^{3+}$. Snapshots of CG model for chain-II **b** and chain-III **c** with six Tb$^{3+}$ ions and eighteen $R$-pempH$^-$ ($R$-pempH$_2$) molecules. To better differentiate the chain-II and chain-III, the bead that can interact with Tb$^{3+}$ is used as pink and lime one, respectively. **d** The initial conformation of the self-assembly system, where the chains-II (and/or chains-III) are distributed uniformly in the simulation box

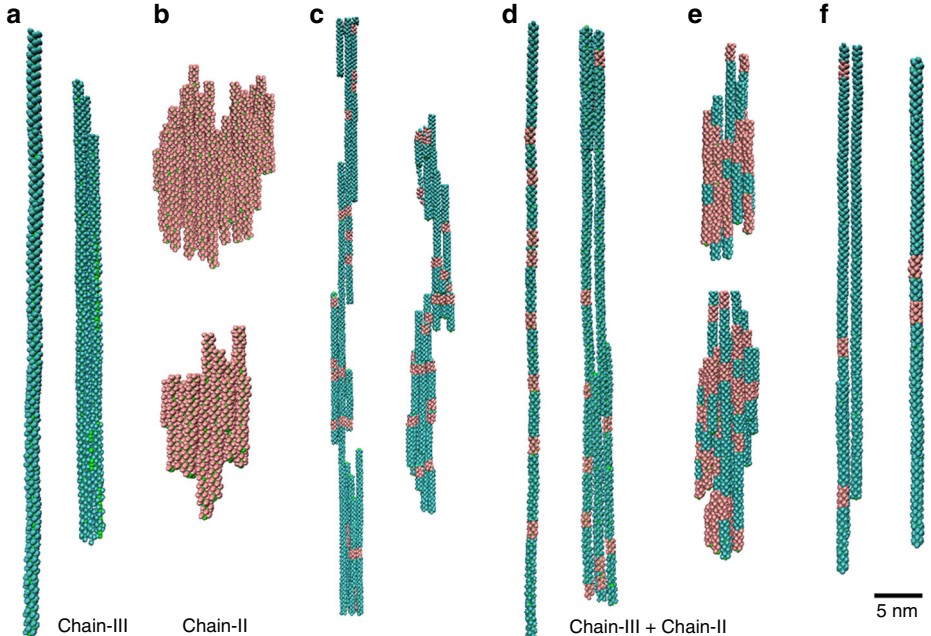

**Fig. 9** Typical snapshots for the assembly in different systems. **a** Pure chain-III system; **b** pure chain-II system; **c** the chain-III and chain-II are mixed as the ratio of 4:1 in the presence of NO$_3^-$; **d** the chain-III and chain-II are mixed as the ratio of 4:1 in the absence of NO$_3^-$; **e** the chain-III and chain-II are mixed as the ratio of 1:1 in the presence of NO$_3^-$; **f** the chain-III and chain-II are mixed as the ratio of 16:1 in the presence of NO$_3^-$. The ochre beads in the $R$-pempH$^-$ / $R$-pempH$_2$ molecules are not shown for clarity

and for regulation of various functions. For example, the DNA triplex formed under acidic conditions will dissociate to form the original duplex under basic conditions[44]. From the molecular level, the impact of slight change in pH is reflected by the different degree of protonation of the phosphonate ligands. As revealed by single crystal structural determination, compound **R-2** formed at lower pH (2.5–2.9) contains two types of positively charged chains, e.g., [Tb$_3$($R$-pempH$_2$)$_2$($R$-pempH)$_7$]$^{2+}$ (chain-I)

and [Tb$_3$($R$-pempH$_2$)($R$-pempH)$_8$]$^+$ (chain-II), where part of the phosphonate groups are singly protonated. Whereas compound **R-3** formed at higher pH (3.3–4.5) contains one type of neutral chain, e.g., Tb($R$-pempH)$_3$ (chain-III), where all phosphonate groups are fully deprotonated. Noting that the positively charged chains in **R-2** are more twisted than the neutral chain in **R-3**, the formation of helices of **R-1** is a natural result of hierarchical self-assembly of both the neutral chains as dominant species and the

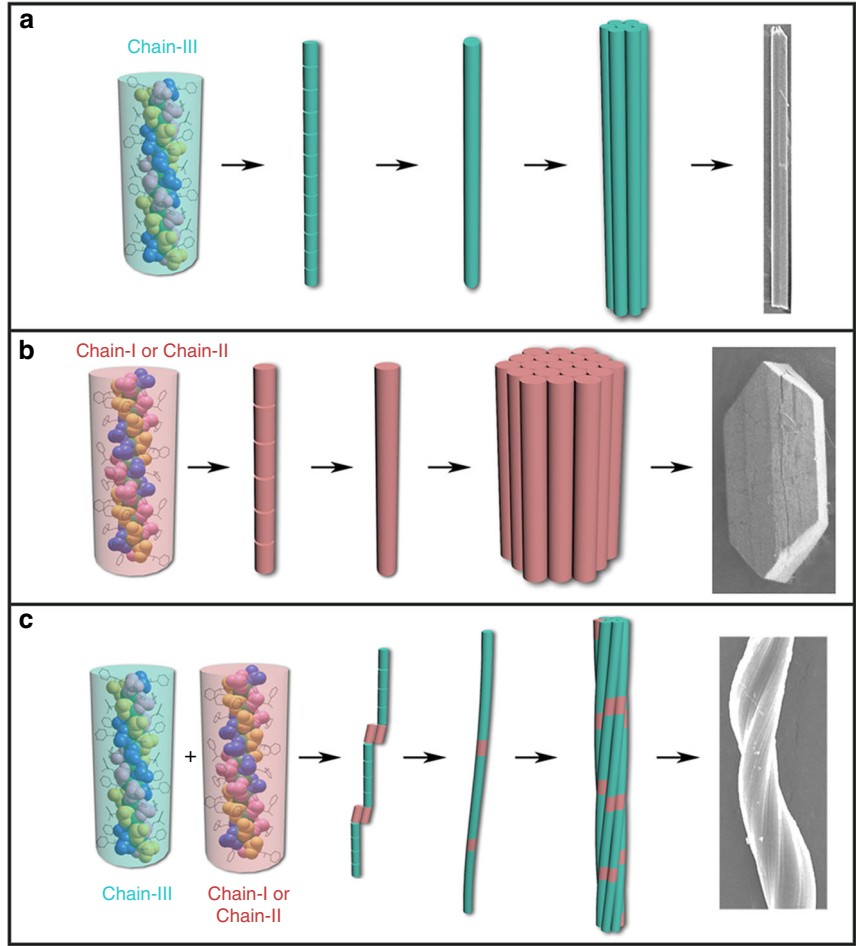

**Fig. 10** Proposed formation mechanism of helices. Formation mechanisms of crystalline materials of **R-3** (**a**) and **R-2** (**b**), and helices of **R-1** (**c**)

positively charged chains, which coexist in the same reaction solution under suitable pH condition (ca. 3.1). The $NO_3^-$ ions, which serve as spacers and hydrogen bond acceptors, are responsible for the twist of the chain aggregates, and hence play an important role in the formation of the helices of **R-1**.

Based on both the experimental and simulation results, the formation mechanism of the helices of **R-1** can be proposed as illustrated in Fig. 10. Assembly starts with the formation of Tb/R-pemp complexes, and fragments of 1D helical chains are constructed via coordination interactions between the $Tb^{3+}$ ions and phosphonate ligands. The degree of protonation of the phosphonate groups and thus the composition of the helical chains in solution are dependent on the pH of the reaction mixture. At pH ca. 3.1, both neutral (chain-III in **R-3**) and positively charged chains (chain-I and chain-II in **R-2**) coexist in the same reaction solution with the former being the dominant species. The binding of the positively charged chains to the neutral ones triggers the twist of chain growth in pure **R-3** system. Since chains of **R-2** prefer to aggregate with each other, the growth along the chain direction becomes more curved and twisted. The twisted chains further assemble into hierarchical bundles, like the artificial supramolecular helices[45,46]. The assembly rate along the length of the helix is faster than that of the width, thus leading to **R-1** with a 1D helical morphology.

The helices of **R-1** are right-handed, whereas the Tb(R-pempH)$_3$ chain (chain-III) has a left-handed helical conformation. Same phenomenon was observed in biomolecules such as collagen, a right-handed helical superhelix composed of three polypeptides where each has a left-handed helical conformation[47]. The chirality of the chains is determined by the stereo configuration of the phosphonate ligand, whereas the transfer of chiral information from molecular to the macroscopic level can be attributed to the weak interchain interactions. Clearly, the pendant amino and phenyl groups of the phosphonate ligands and nitrate ions play non-trivial roles in the successful construction of helices of **R-1** and **S-1**.

In summary, we report homochiral CPs with formulae [Tb(R- or S-pempH)$_3$]•2H$_2$O with both helical and crystalline forms. The geometric incompatibility of the coexisting chain types in the same reaction mixture leads to the formation of helices. This work provides new insight into the design and construction of homochiral CPs with helical morphologies. Furthermore, the observed chiral transcription from molecules to morphologies is also significant for understanding bio-related self-assembly processes on the nano- to macro-scale.

## Methods

**Materials and physical measurements.** R- or S-(1-phenylethylamino)methyl-phosphonic acid (pempH$_2$) was prepared according to methods reported in the literature[38]. All other starting materials were of reagent grade and were used as received from commercial sources without further purification. Elemental analyses for C, H and N were carried out on a PE 240C analyser. IR spectra were recorded with a Bruker Tensor 27 spectrometer using KBr discs. The pH value was measured by a Sartorius PB-10 pH metre. Thermal analyses were performed under nitrogen in the temperature range of 25–800 °C at a heating rate of 5 °C min$^{-1}$ on a METTLER TOLEDO TGA/DSC 1 instrument. PXRD data were collected using a Bruker D8 advance diffractometer. SEM measurements were performed on SHI-MADZU SSX-550. The UV–Vis spectra were recorded on Perkin Elmer Lambda

950 UV–Vis/NIR Spectrometer. The circular dichroism spectra were recorded on a JASCO J-720 W spectropolarimeter at room temperature. The simultaneous CD and LD measurements were conducted for *R*-1 and *S*-1 using the multi-probe function in the J-1500 CD spectrometer. Each sample was diluted by KCl with a ratio of 1/50 and pressed into a pellet.

**Preparation of h-Tb($R$-pempH)$_3$•2H$_2$O ($R$-1) helices**. A mixture of Tb (NO$_3$)$_3$•6H$_2$O (0.1 mmol, 0.0468 g) and $R$-pempH$_2$ (0.5 mmol, 0.1080 g) in 6 mL of H$_2$O was stirred for 2 h at room temperature, and then the pH of the mixture was adjusted to pH 3.1 with 0.5 M NaOH. Afterwards, the glass containing the mixture was kept in a Teflon-lined autoclave (15 mL) with additional 4 mL water, and allowed for hydrothermal reactions at 120 °C for 2 d. After cooling to room temperature, the flocculent precipitates of $R$-1 helices were collected manually and dried under air. Elemental analyses calcd (%) for C$_{27}$H$_{39}$N$_3$O$_9$P$_3$Tb•2H$_2$O: C 38.72, H 5.17, N 5.02; found: C 38.67, H 5.25, N 4.98. IR (KBr): $\bar{v} = 3409$ (s), 3048 (m), 2985 (m), 2784 (m), 2518 (m), 2404 (w), 1620 (m), 1492 (m), 1457 (m), 1423 (w), 1382 (w), 1273 (m), 1149 (m), 1079 (s), 1022 (s), 987 (s), 765 (m), 701 (m), 567 (m), 537 (m), 509 (m), 476 (m) cm$^{-1}$. Thermal analysis revealed that the weight loss below 100 °C was 4.4%, in agreement with the release of two lattice water molecules (calcd. 4.3%).

**Preparation of h-Tb($S$-pempH)$_3$•2H$_2$O ($S$-1) helices**. $S$-1 was obtained using the same procedure as for $R$-1 except that $S$-pempH$_2$ was used as the starting material. Elemental analyses calcd (%) for C$_{27}$H$_{39}$N$_3$O$_9$P$_3$Tb•2H$_2$O: C 38.72, H 5.17, N 5.02; found: C 38.73, H 5.27, N 4.91. IR (KBr): $\bar{v} = 3409$ (s), 3048 (m), 2985 (m), 2784 (m), 2518 (m), 2404 (w), 1620 (m), 1492 (m), 1457 (m), 1423 (w), 1382 (w), 1273 (m), 1149 (s), 1079 (s), 1022 (s), 987 (s), 765 (m), 701 (m), 567 (m), 536 (m), 509 (m), 475 (m) cm$^{-1}$. Thermal analysis revealed that the weight loss below 100 °C was 4.4%, in agreement with the release of two lattice water molecules (calcd. 4.3%).

**Preparation of block-like crystals of $R$-2**. Compound $R$-2 was obtained following the same procedure as for $R$-1 except that the pH of the mixture was adjusted to 2.7. Colourless needle-like crystals were obtained. Elemental analysis calcd (%) for C$_{81}$H$_{119}$N$_{11}$O$_{33}$P$_9$Tb$_3$•5H$_2$O: C 37.09, H 4.92, N 5.88; found: C 37.24, H 4.81, N 5.79. IR (KBr): $\bar{v} = 3423$(s), 3066(w), 2983(w), 2794(w), 2520(w), 2079(w), 1622 (m), 1458(m), 1384(m), 1274(w), 1143(s), 1120(s), 1085(s), 1035(s), 989(s), 931(w), 766 (m), 702 (m), 665(w), 628(w), 601(w), 563 (m), 538 (m), 507(m), 474 (m) cm$^{-1}$. Thermal analysis revealed that the weight loss below 100 °C was 3.5%, in agreement with the release of five water molecules per Tb$_3$ (calcd. 3.4%).

**Preparation of block-like crystals of $S$-2**. Crystals of $S$-2 were obtained using the same procedure as for $R$-2 except that $S$-pempH$_2$ was used as the starting material. Elemental analysis calcd (%) for C$_{81}$H$_{119}$N$_{11}$O$_{33}$P$_9$Tb$_3$•5H$_2$O: C 37.09, H 4.92, N 5.88; found: C 37.30, H 4.84, N 5.74. IR (KBr): $\bar{v} = 3418$(s), 3065(w), 2987(w), 2799 (w), 2524(w), 2075(w), 1616(m), 1458(m), 1384(m), 1274(w), 1143(s), 1120(s), 1085(s), 1035(s), 987(s), 930(w), 766 (m), 702 (m), 665(w), 628(w), 601(w), 565 (m), 536 (m), 505(m), 474 (m) cm$^{-1}$. Thermal analysis revealed that the weight loss below 100 °C was 3.4%, in agreement with the release of five water molecules per Tb$_3$ (calcd. 3.4%).

**Preparation of rod-like crystals of $R$-3**. Compound $R$-3 was obtained following the same procedure as for $R$-1 except that the pH of the mixture was adjusted to 3.7. Colourless needle-like crystals were obtained. Elemental analysis calcd (%) for C$_{27}$H$_{39}$N$_3$O$_9$P$_3$Tb•2H$_2$O: C 38.72, H 5.17, N 5.02; found: C 38.92, H 5.24, N 5.09. IR (KBr): $\bar{v} = 3409$ (s), 3048 (m), 2985 (m), 2784 (m), 2518 (m), 2404 (w), 1620 (m), 1492 (m), 1457 (m), 1423 (w), 1382 (w), 1273 (m), 1149 (s), 1079 (s), 1022 (s), 987 (s), 765 (m), 701 (m), 567 (m), 536 (m), 509(m), 476 (m) cm$^{-1}$. Thermal analysis revealed that the weight loss below 100 °C was 3.8%, in agreement with the release of two lattice water molecules (calcd. 4.3%).

**Preparation of rod-like crystals of $S$-3**. Crystals of $S$-3 were obtained using the same procedure as for $R$-3 except that $S$-pempH$_2$ was used as the starting material. Elemental analysis calcd (%) for C$_{27}$H$_{39}$N$_3$O$_9$P$_3$Tb•2H$_2$O: C 38.72, H 5.17, N 5.02; found: C 38.99, H 5.24, N 4.93. IR (KBr): $\bar{v} = 3409$ (s), 3048 (m), 2985 (m), 2784 (m), 2518 (m), 2404 (w), 1620 (m), 1492 (m), 1457 (m), 1423 (w), 1382 (w), 1273 (m), 1149 (s), 1079 (s), 1022 (s), 987 (s), 765 (m), 701 (m), 567 (m), 536 (m), 509 (m), 476 (m) cm$^{-1}$. Thermal analysis revealed that the weight loss below 100 °C was 3.8%, in agreement with the release of two lattice water molecules (calcd. 4.3%).

**Determination of crystal structures**. Single crystals of dimensions 0.25 × 0.25 × 0.20 mm$^3$ for $R$-2, 0.30 × 0.20 × 0.15 mm$^3$ for $R$-2′, 0.40 × 0.10 × 0.10 mm$^3$ for $R$-3, and 0.30 × 0.05 × 0.05 mm$^3$ for $S$-3 were used for data collection on a Bruker APEX DUO (for $R$-2 and $R$-2′) or D8 (for $R$-3 and $S$-3) diffractometer using graphite-monochromated Mo Kα radiation ($\lambda = 0.71073$ Å) at 150 K. To confirm the position of the lattice water molecules, another group of single crystals of dimensions 0.30 × 0.10 × 0.10 mm$^3$ for $R$-3 and 0.60 × 0.15 × 0.15 mm$^3$ for $S$-3 was

selected and sealed in the mother solution for data collection on a Bruker D8 diffractometer using graphite-monochromated Mo Kα radiation ($\lambda = 0.71073$ Å) at 277 K. The numbers of collected and observed independent $[I > 2\sigma(I)]$ reflections were 83656 and 24907 ($R_{int} = 0.150$) for $R$-2, 80065 and 17686 ($R_{int} = 0.079$) for $R$-2′, 40159 and 5473 ($R_{int} = 0.067$) for $R$-3, 39847 and 5705 ($R_{int} = 0.065$) for $S$-3, 39562 and 5130 ($R_{int} = 0.071$) for $R$-3 (277 K), and 42447 and 5763 ($R_{int} = 0.040$) for $S$-3 (277 K). The data were integrated using the Siemens SAINT programme[48]. Adsorption corrections were applied. The structures were solved using direct methods and refined on $F^2$ using full-matrix least-squares using SHELXTL[49]. Anisotropic temperature factors were used to refine all atoms excluding hydrogen. All hydrogen atoms bound to carbon were refined isotropically in riding mode; hydrogen atoms of water molecules were detected via experimental electron density and then refined isotropically with reasonable restriction of O–H bond distances and H–O–H angles. For $R$-3 and $S$-3, only one lattice water molecule with 0.5 occupancy could be determined from difference Fourier maps due to poor diffraction data or lattice solvent disorder even though the samples were sealed in mother solution for data collection. And the number of water molecules in lattice was determined by thermal analyses and elemental analyses. The crystallographic data are given in Supplementary Table 3, and the selected bond lengths and angles are shown in Supplementary Tables 4, 6, 8 and 11.

**All-atom molecular dynamics simulations**. The all-atom MD simulations were performed by using Gromacs 5.0.4 package[50] with the Amber force field[51] and the TIP3P water model[52] in the NVT ensemble. The force field parameter for $R$-pempH$^-$ and $R$-pempH$_2$ was built by using Antechamber tool[53]. The parameter for Tb$^{3+}$ was adopt by Li et al.[54] and that for NO$_3^-$ was adopted by Liu et al.[55] During the simulation, the temperature was coupled at 300 K using Nosé-Hoover method[56]. The Particle Mesh Ewald (PME) method was used to calculate the electrostatic interactions and the cutoff of Lennard–Jones (LJ) interaction was 1.2 nm. The periodic boundary conditions were applied in all three dimensions. The time step was chosen as 2 fs and each simulation was at least conducted for 50 ns.

**Brownian dynamics simulations**. In order to better understand the distinct assembly under different cases, the BD simulations[57] were also used. Actually, since there existed a large number of chain-II and/or chain-III in this system, it was far beyond the computing ability of present all-atom molecular simulation. Here, the Tb$^{3+}$ was treated as one (green) bead (i.e., Tb bead) and the $R$-pempH$^-$/$R$-pempH$_2$ molecule was simplified as two beads, where the ochre bead (i.e., P1 bead) can interact with each other to promote the growth along the side direction and the pink (for chain-II) or lime (for chain-III) bead (i.e., P2 bead) can interact with Tb bead to promote the growth along the chain direction (Fig. 9a). The harmonic spring interaction $U_s = k_s(l_{i,i+1} - l_0)^2$ was applied between connected beads in the polymers and the binding sites of particle beads, where $k_s = 3000 \, k_B T_0 \, r_0^{-2}$, $l_0 = 0.4 \, r_0$. In order to depict the attractive interaction of the Tb bead-P2 bead, the P1 bead-P1 bead, the Lennard–Jones potential was used ($\sigma = 0.4 \, r_0$). According to the all-atom simulation, the interaction energy between Tb$^{3+}$ and $R$-pempH$^-$ was fifteen times as large as that between $R$-pempH$^-$ and $R$-pempH$^-$ ($R$-pempH$_2$) molecules. Here for the sake of simplicity, we set $\epsilon_{p1-p1}/k_B T_0 = 1.0$ and $\epsilon_{Tb-p2}/k_B T_0 = 15.0$. In particular, when in the presence of NO$_3^-$, the all-atom simulation results showed that the interaction energy between $R$-pempH$^-$ and $R$-pempH$_2$ molecules was three times as large as that in the absence of NO$_3^-$. Thus, we set the $\epsilon_{p1-p1}/k_B T_0 = 3.0$ in chain-II when in the presence of NO$_3^-$. Besides, the shifted Lennard–Jones potential, cutoff at $2^{1/6} \, \sigma$, was used to model the repulsive interaction between the other beads ($\epsilon/k_B T_0 = 1.0$, $\sigma = 0.4 \, r_0$). Since there are six Tb$^{3+}$ ions and eighteen $R$-pempH$^-$/$R$-pempH$_2$ molecules in each pitch of chain-II/chain-III, the monomer for the assembly in the simulations was chosen as six Tb beads with eighteen ochre beads and 18 pink or lime beads (see Fig. 9b, c). In addition, the initial coordinate of Tb beads and P1 beads (in the monomer) was obtained by using the coordinate of Tb$^{3+}$ ion and P atom (of $R$-pempH$^-$ molecule) in experimental crystal data, respectively. Moreover, Tb beads and P1 beads in each monomer were treated as rigid ones in BD simulation, namely, the relative position of Tb beads and P1 beads in the monomer was kept fixed, which was due to the following two reasons. First, as shown in Supplementary Fig. 42, the all-atom MD simulation results showed that the relative position of Tb$^{3+}$ ion and P atom changed very little since the peak of the Tb–P distance and the P–Tb–P angle was very sharp in the probability distribution functions. Actually, the values of these parameters in all-atom MD simulations were close to those in the experiments (see the caption in Supplementary Fig. 42). Second, if we did not constrain the relative position of Tb beads and P1 beads in the monomer, the molecular structure obtained in BD simulation could be totally different from that in the all-atom MD simulation, and more importantly, differed from the unit in the experimental structure (e.g., chain-II and chain-III cannot be distinguished). As a result, here we treat the Tb beads and P1 beads in each monomer as rigid ones in the BD simulations.

All BD simulations were performed in the NVT ensemble by using the LAMMPS package (15 May 2015)[58]. During the simulation, as did in the experiments, the temperature was firstly coupled at 1.3 $T_0$ (about 390 K or 120 °C) in the initial 100,000 τ and then decreased to 1.0 $T_0$ (about 300 K, i.e., room temperature) using the Langevin thermostat. The time step was 0.008 τ, and the data were collected every 80 τ, with the total simulation time lager than 400,000 τ.

**Data availability**. The X-ray crystallographic coordinates for structures reported in this study have been deposited at the Cambridge Crystallographic Data Centre (CCDC), under deposition numbers 1501001–1501006. These data can be obtained free of charge from The Cambridge Crystallographic Data Centre via "http://www.ccdc.cam.ac.uk/data_request/cif". Other data that support the findings of this study are available on request from the corresponding authors.

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

## Acknowledgements

This work is supported by the Ministry of Science and Technology of China (2017YFA0303200), the National Natural Science Foundation of China (21371094, 91427302 and U1532110) and the National Basic Research Programme of China (2013CB922102). We thank Prof. Feng Li in Zhengzhou University of Light Industry, Prof. Hong-Zhen Lian in Nanjing University and Prof. Shunai Che in Shanghai Jiaotong University for valuable helps and discussions.

## Author contributions

L.M.Z. and J.H. initiated the study. J.H. synthesised the helical materials and characterised the compounds. D.Z. repeated the experiments and isolated the *R*-**2** and *R*-**2′** phases. Y.X. carried out the systematic studies on the formation mechanism of the helices. H.Z. and D.M.Z. obtained the SEM images. S.S.B. analysed the single crystal

structures. H.D. and Y.M. performed the molecular simulations. J.H., S.S.B., H.D., Y.M. and L.M.Z. interpreted the data and wrote the manuscript. All authors discussed the results and contributed to the manuscript.

## Additional information

**Competing interests:** The authors declare no competing financial interests.

