## [Peer Review File · Nature Communications]

Reviewers' comments:

Reviewer #1 (Remarks to the Author):

The manuscript reports a family of new chiral coordination framework materials consisting of 1-D helical chains of Tb(III) bridged by two different chiral ligands. These compounds are of some general interest in adding to the rapidly growing family of chiral framework structures, but in many ways are themselves structurally unremarkable. Whilst the materials are in many ways themselves structurally unremarkable, the crucial point of novelty of the work is the observation that self-assembly of different types of these 1-D chains can lead to chirality on the macroscale in the form of helical macrostructures. Such expression of chirality from the molecular scale to the macroscale appears to be relatively rare outside of biological systems (and beyond that present in Pasteur's crystal morphology method for chiral separation, which is quite a separate effect), with the closest examples to that reported here seemingly being references 21 and 22, which provide examples of helical nanobelts. Whilst the observation of helical macrostructures appears potentially to be of some considerable interest, there are a number of deficiencies with the work and its presentation that make it difficult to assess the degree of novelty and significance; overall, my impression is that this manuscript is not quite detailed enough, nor is the contribution suitably described and positioned with respect to other work within the field, to justify publication in Nature Comm.

My specific points are as follows:

As far as I can tell (noting that I have not found this to be a straightforward area to search the literature on) the mechanism suggested is new: that the co-crystallisation of two or three slightly different chiral chains causes the formation of the chiral morphology. However, there really needs to be more supporting evidence, either experimental (showing more than one type of chain present in the crystal) or modelling (like that in ref 21) to confirm the mechanism of the helicate formation. A range of diffraction-based techniques (X-rays and electrons) could have been employed to identify the nature and causes of the macro-chirality (as was done in refs 21 and 22). Additionally, there is scope for further microscopy of the materials during crystal growth. In ref 21, describing the growth of amino acid 'bio coordination polymers', there is a description of the growth of the particles (from SEM of samples taken during synthesis), with non-helical fibres growing into helical belts. (They claim that the "nonhelical belts continued assembling and merging with each other through lateral attachment", although I can't see the evidence for this mechanism.)

Since the proposed mechanism is important for this manuscript, the introduction should not only list previous examples of chiral macrostructures, but also examine the proposed mechanisms by which these morphologies are formed from chiral molecular components. Comparing refs 21 & 22, the mechanisms suggested seem to be quite varied and ill-defined.

Much of the detail given in the manuscript is peripheral to what I regard as the core point of importance – that of the generation of macroscale chirality. The introduction is devoted largely to a description of the general class of materials and the results and discussion are devoted largely to description of the crystal structures. Overall, the feeling as a reader is that there is something very interesting going on here, but then disappointment that the work hasn't progressed to a level where a more detailed and definitive mechanism can be determined and then contrasted with other instances of this type of effect.

Figure 6 should make clear which of the chains shown are Chains-I, -II or -III.

Reviewer #2 (Remarks to the Author):

Understanding the transfer or amplification of chiral information within different scales, such as

from molecular to macroscopic level, still remains a topic of great interest and activity. For coordination polymers, fabrication of chiral assemblies with helical structures taking the form of both single crystal and microscale architectures is difficult. This paper demonstrates the chirality transfer of artificial coordination polymers with very nice single crystal data, which might have chance to be published in nature communication. However, the following issues must be addressed before the final determination for acceptance.

(1) The formation of single crystals with helical morphologies, especially the impact from pH values and anions, has been clearly discussed in the manuscript. The mechanism concerning chirality transfer from molecular assembly to micrometer scale, however, seems to be still not clear and lack of enough experimental evidence. For example, the manuscript shows the SEM images of the microscale helixes with the diameters of about 6-9 μm and the helical crystal structures with the diameters less than 2 nm. However, the morphologies of the assemblies around nanoscales were not investigated. As a result, there appears no enough obvious evidence for supporting the formation of hierarchical bundles from molecular scales to nanoscales. I suggest the authors to perform the TEM measurement for these systems, which might provide more clear and useful information in this regard.

(2) For the solid-state samples, circular dichroism (CD) spectral measurement should be cautious. The LD artifacts should be considered. Please provide details of the experimental conditions of corresponding spectral measurements.

(3) In addition, more detailed experimental conditions should be provided. For example, the pH value of the systems should be measured both before and after hydrothermal reactions.

Reviewer #3 (Remarks to the Author):

This is an interesting manuscript reporting on the synthesis and characterization of a chiral terbium-based coordination polymer. The work is sound and well reported. I think it is suitable for publication in Nature Communications after some modifications, to improve the understanding of the system. Right now, this is rather a technical report, with all the data gathered but not much in terms of physical and chemical understanding of the system.

I would suggest three improvements to the work before publication:

- The visualization of the coordination (e.g. Figure 3a) is rather poor, and I have a very hard time understanding the nature of the coordination pattern around the metal cations.
- The formation of "self-twisting bundles" (Figure 6 and related text) is not convincingly backed by data. Right now, it sounds more like a hypothesis of the authors. I suggest they gather data to prove this hypothesis. Especially as it features heavily in the manuscript, and in the graphical abstract too.
- More chemical and physical analysis of the system needs to be given. Right now, the study convincingly shows the formation of this chiral CP, but does not explain to the reader why it is rare and specific to this particular system. Maybe an analysis of the packing or contact surfaces, or computations, could give a better physical understanding of the system?

Response to the referees' questions:

For Reviewer #1:

The manuscript reports a family of new chiral coordination framework materials consisting of 1-D helical chains of Tb(III) bridged by two different chiral ligands. These compounds are of some general interest in adding to the rapidly growing family of chiral framework structures, but in many ways are themselves structurally unremarkable. Whilst the materials are in many ways themselves structurally unremarkable, the crucial point of novelty of the work is the observation that self-assembly of different types of these 1-D chains can lead to chirality on the macroscale in the form of helical macrostructures. Such expression of chirality from the molecular scale to the macroscale appears to be relatively rare outside of biological systems (and beyond that present in Pasteur's crystal morphology method for chiral separation, which is quite a separate effect), with the closest examples to that reported here seemingly being references 21 and 22, which provide examples of helical nanobelts. Whilst the observation of helical macrostructures appears potentially to be of some considerable interest, there are a number of deficiencies with the work and its presentation that make it difficult to assess the degree of novelty and significance; overall, my impression is that this manuscript is not quite detailed enough, nor is the contribution suitably described and positioned with respect to other work within the field, to justify publication in Nature Comm.

Reply: We appreciate the reviewer for considering the work as "potentially of some considerable interest".

(1) As far as I can tell (noting that I have not found this to be a straightforward area to search the literature on) the mechanism suggested is new: that the co-crystallisation of two or three slightly different chiral chains causes the formation of the chiral morphology. However, there really needs to be more supporting evidence, either experimental (showing more than one type of chain present in the crystal) or modelling (like that in ref 21) to confirm the mechanism of the helicate formation. A range of diffraction-based techniques (X-rays and electrons) could have been employed to identify the nature and causes of the macro-chirality (as was done in refs 21 and 22). Additionally, there is scope for further microscopy of the materials during crystal growth. In ref 21, describing the growth of amino acid 'bio coordination polymers', there is a description of the growth of the particles (from SEM of samples taken during synthesis), with non-helical fibres growing into helical belts. (They claim that the "nonhelical belts continued assembling and merging with each other through lateral attachment", although I can't see the evidence for this mechanism.)

Reply: We appreciate the reviewer for raising these critical questions. We agree that more evidence is needed to support the proposed mechanism of helix formation. According to your suggestion, we have performed more experiments and also used molecular simulations to reveal the underlying mechanism of helices formation.

Experimental part:

In the previous version of the manuscript, we presented the SEM images of the products after hydrothermal reactions of $\text{Tb}(\text{NO}_3)_3$ and $R\text{-pempH}_2$ at pH 3.1 and 120 °C for 1-24 h. The product obtained after 1 h had already taken the form of helical nanofibrils with different diameters and lengths. In order to get more insights into the formation mechanism of the chiral morphology, we repeated the experiments, and monitored the solid products after hydrothermal reactions for different period of time (0 min, 5 min, 10 min, 15 min, 20 min, 30 min, 1 h, 2 h, 4 h, 6 h, 8 h, 10 h, 20 h, and 48 h) by using PXRD, IR and SEM measurements. The results provide clear evidence supporting our proposed mechanism of helix formation, e.g. the co-crystallisation of two or three slightly different chiral chains causes the formation of the chiral morphology.

The experiments were carried out by stirring a suspension of $\text{Tb}(\text{NO}_3)_3$ and $R\text{-pempH}_2$ in water (6 mL) for 2 hours at room temperature, and then adjusting the mixture to pH 3.1 by 0.5 M NaOH. Afterwards, the glass containing the mixture was put into a Teflon lined autoclave (15 mL) with additional 4 mL water, and allowed for hydrothermal reactions at 120 °C for different period of time. Upon cooling to room temperature, the solid products were filtered and washed with water. The product of 0-min referred to the collected precipitate right before the hydrothermal reaction was conducted.

Figure R1 (Supplementary Figure 31 in the new version) shows the PXRD patterns of the hydrothermal reaction products. Those of pure $R\text{-pempH}_2$, **R-2** and **R-3** are also given for a comparison. It is clear that the 0-min product contained only the ligand of $R\text{-pempH}_2$. After hydrothermal reaction for 10 min, helices of **R-1** appeared together with the ligand, as evidenced by the emergence of a peak at a lower angle of 6.183° . The two phases, e.g. **R-1** and the solid $R\text{-pempH}_2$, coexisted in the reaction mixture even after hydrothermal reaction for 1 h. When the reaction time reached 2 h and above, pure phases of **R-1** can be observed. The results suggested that the formation of the helices of **R-1** was quite fast (within 10 min) under hydrothermal conditions at pH 3.1 and 120 °C. The crystalline materials of **R-2** and **R-3** were not found in the final products of different reaction times (0 - 48 h).

Figure R1. Top: The PXRD patterns of the products after hydrothermal reactions of $\text{Tb}(\text{NO}_3)_3$ and $R\text{-pempH}_2$ (pH \sim 3.1) at 120 °C for different period of time. The PXRD patterns of the ligand, the block-like crystals of **R-2** and the rod-like crystals of **R-3** is also given for a comparison. Bottom: Enlarged PXRD patterns of the same products for clarity.

Considering that both the chemical composition and the structure of **R-1** are close to those of **R-3** which contains neutral chains of chain-III, the PXRD measurements cannot help supporting our proposal that the involvement of chains of **R-2** is important for the formation of the helices of **R-1**. Since the chains in **R-2** are positively charged, which should be charge-balanced by the NO_3^- anion. Thus the detection of the nitrate anion in the final products by IR spectra may give us some clue of the formation mechanism.

Figure R2 (Supplementary Figure 32 in the new version) shows the IR spectra of the above-mentioned products. The presence of un-coordinated nitrate anions can be identified by the appearance of a peak at ca. 1385 cm^{-1} . As expected, this peak was significant in the case of compound **R-2**. For the hydrothermal reaction products of $\text{Tb}(\text{NO}_3)_3$ and $R\text{-pempH}_2$,

the peak intensity at ca. 1385 cm^{-1} was markedly increased when the reaction time reached 4-6 h, indicating that the NO_3^- anion and hence the positively charged chains (chain-I, -II) could be indeed involved in the helices of **R-1**. The increase of the peak intensity cannot be identified when the reaction time was less than 2 h or more than 8 h, possibly due to the interference of the ligand and/or the presence of neutral chains in the helices of **R-1**. The results suggested that the amount of positively charged chains in the helices of **R-1** could be very small compared with that of the neutral ones. This may explain the fact that both the PXRD pattern and the chemical composition of the helices of **R-1** are close to those of compound **R-3**.

Figure R2. The IR spectra in the ranges of $540\text{-}4000\text{ cm}^{-1}$ (a) and $540\text{-}2100\text{ cm}^{-1}$ (b) of the products after hydrothermal reactions of $\text{Tb}(\text{NO}_3)_3$ and $R\text{-pempH}_2$ ($\text{pH} \sim 3.1$) at $120\text{ }^\circ\text{C}$ for different period of time. The IR spectra of the ligand, the rod-like crystals of **R-3** and the block-like crystals of **R-2** are also given for a comparison. (c) The variation of the peak intensity at 1385 cm^{-1} for the time-dependent products. The peak at 702 cm^{-1} was assigned to

the vibration of mono-substituted phenyl, which was used as the internal standard to eliminate the differences caused by the concentrations of different samples. (d) The normalized spectra with the baseline subtracted, from which the peak intensity at 1385 cm^{-1} was abstracted.

Figure R3. SEM images and EDX analysis of the 0-min product in different regions.

Figure R4. EDX analysis of *R*-pempH₂, after stirring in 6 mL water for 2 hours.

Although the PXRD measurements suggested the formation of the helices of **R-1** after hydrothermal reactions of $\text{Tb}(\text{NO}_3)_3$ and $R\text{-pempH}_2$ (pH \sim 3.1) at 120 °C for 10 min, the SEM images can give a clear visualization about the morphology of the products. Figure R3 (top) (Supplementary Figure 33 in the new version) shows the SEM images of 0-min products. Before treatment under hydrothermal conditions, the 0-min product contained only sheets covered by some amorphous nano-particles (size: 100-300 nm). The EDX analyses revealed that there are more Tb atoms present in the nano-particles (Tb/P ratio: 1/15) than that in the smooth area without nano-aggregates (Tb/P: 1/72) (Figure R3, bottom). In contrast, no Tb atom was detected by EDX in the pure ligand of $R\text{-pempH}_2$ after stirring in 6 mL water for 2 hours (Figure R4, Supplementary Figure 34 in the new version). The results suggested that the 0-min product contained the un-dissolved ligand of $R\text{-pempH}_2$ with the presence of a small amount of amorphous terbium complexes which cannot be identified by PXRD measurements.

Figure R5 (Supplementary Figure 35 in the new version) shows the SEM images of the 15-min product. Obviously, the solid sheet was covered by aggregates of both nano-particles and nanorods. Although the nanorod did not show clear helical morphologies, the atomic ratio of Tb/P was close to 1/3. In contrast, the Tb/P ratio of the nano-particle area is ca. 1/7 (Figure R6, Supplementary Figure 35 in the new version). The result indicated that the nano-particles on the un-dissolved ligand of $R\text{-pempH}_2$ can further react with terbium ions in solution, and formed nanorods which composition is close to the helices of **R-1**.

Figure R5. SEM images of the 15-min product.

Figure R6. EDX analysis of the 15-min product in different regions.

Figure R7. SEM images (top) and EDX analyses (bottom) of the 20-min product.

Interestingly, small helices are clearly observed in the product of 20-min like an actinian, the widths and lengths of which are ca. 150-500 nm and 2-5 μm , respectively (Figure R7, top, Supplementary Figure 36 in the new version). The growth of the helices seemed to start on the surface of the un-dissolved ligand of *R*-pempH₂, which was supported by the EDX analysis. As shown in Figure R7 (bottom), the Tb/P atomic ratio was ca. 1:3 for the helices, but ca. 0/3 for the bottom area without helices. Combined with the results of 15-min products, we suppose that the nano-particles formed first on the surface of the un-dissolved ligand of *R*-pempH₂. Then the nano-particles further reacted with the terbium ions to form nanorods. Finally the nanorods grew into helices of ***R*-1**.

Figure R8. SEM images of the 30-min products.

After reacting for 30 min under hydrothermal conditions, aggregates of helices can be observed in the 30-min product, which looks like the bird's nest (Figure R8, Supplementary Figure 37 in the new version). Compared with the 20-min product, however, the lengths of the helices (ca. 2.05 - 11.78 μm) became longer, and the widths (ca. 0.19 - 0.97 μm) became wider. For the 1-h product, starfish-like aggregates of helices can be recognized together with the separate ones (Figure R9, Supplementary Figure 38 in the new version). Meanwhile, the lengths (ca. 6.37 - 119.45 μm) and the widths (ca. 0.19 - 2.26 μm) of the helices increased further, from nanometer to micrometer scale. After 2-8 hours of hydrothermal reactions, the helices of ***R*-1** appeared separately, and the average length and width of the helices increased with increasing reaction time (Figures R9, R10, Supplementary Figures 38 and 39 in the new version). Notably, the widths of the helices did not change significantly

after 8 hours of reactions, although the lengths of the helices increased continuously (Figure R10).

Figure R9. SEM images of the 1-h and 2-h products.

Figure R10. SEM images of the 6-h, 8-h, 10-h and 20-h products.

We found that the presence of undissolved solid ligand of *R*-pempH₂ was important for the formation of helices of **R-1**, because the growth of helices occurred on the surface of the solid ligand. The role of solid ligand of *R*-pempH₂ could be two fold: (1) it serves as a buffer against the pH change in solution during the self-assembly process. The pH of a saturated solution of *R*-pempH₂ is ca. 3.5 (Figure R11, Supplementary Figure 40 in the new version). The coordination of *R*-pempH₂ with Tb³⁺ would release the protons, and then decrease the pH in solution. Thus, the dissolve of solid *R*-pempH₂ could help maintaining the pH of solution in a suitable range for helix formation. (2) it serves as nucleation centers at which the helices of **R-1** grow, as observed in the SEM images of the 20-min and 30-min products.

Figure R11. The pH values of the saturated solutions of *R*-pempH₂ (10 mg/mL) measured for ten parallel batches.

Considering that the temperature could affect the growth rate of materials, we conducted the same reactions (pH 3.1) under hydrothermal conditions at temperatures 80-160°C for 20 h. As shown in Figure R12 (Supplementary Figure 17 in the new version), the 80°C product gave only particles on the surface of the undissolved ligand. When the reaction temperature reached 100-120°C, helices of **R-1** were obtained. The helices can also be found in the 140°C product, but the pitches of the helices were significantly larger. No obvious helices can be found in the 160°C product. The results suggested that the lengths, widths and pitches of the helices of **R-1** were highly dependent on the temperature.

Figure R12. SEM images of the products after hydrothermal reactions of $\text{Tb}(\text{NO}_3)_3$ and $R\text{-pempH}_2$ with a molar ratio of 1:5 (pH 3.1) at different temperatures for 20 h.

From the above experimental results, we can conclude that the helices of **R-1** formed on the surface of the un-dissolved ligand of *R*-pempH₂, forming nano-particles first, then the nanorods, and finally the helices of **R-1**. The growth of the helices followed a hierarchical process with the length direction growing much faster than the width direction. Both the neutral chains of **R-3** and the positively charged chains of **R-2** are involved in the helix formation process.

Simulation part:

But what is the driving force leading to the twist of the chains during the self-assembly process? And why do slightly different chains show distinct assembly forms? To well explain these puzzled problems, we then applied the all-atom molecular dynamics simulation to investigate the interaction energy among Tb³⁺, *R*-pempH⁻, *R*-pempH₂, and also used the coarse-grained (i.e., Brownian dynamics) simulation to study the self-assembly of chain-II and chain-III under different cases.

Figure R13. The setup and result of all-atom molecular dynamics simulations. The initial conformation for different systems (the coordinates are adopted from the experimental crystal data): (a) two *R*-pempH⁻ molecules; (b) one *R*-pempH⁻ molecule and one *R*-pempH₂ molecule; (c) one *R*-pempH⁻ molecule and one *R*-pempH₂ molecule in the presence of NO₃⁻; (d) one Tb³⁺ and three *R*-pempH⁻ molecules. The water molecules and counterions are not shown for clarity. (e) Interaction energy among different molecules in the above systems.

As shown in Figure R13 (Figure 7 in the new version), the interaction energy between Tb³⁺ and *R*-pempH⁻ was much larger (about fifteen times) than that between *R*-pempH⁻ and *R*-pempH⁻ (or *R*-pempH₂), indicating that the growth rate along the chain direction should be faster than that along the side direction. However, in the presence of NO₃⁻, since there existed the hydrogen bonds among the NO₃⁻, *R*-pempH⁻, and *R*-pempH₂, the interaction

energy increased a lot (about three times). As a result, the difference of the growth rate between the chain direction and side direction would not become very obvious and the length and diameter of assembly would also be comparable, which may help explain the block crystal in **R-2** system.

Figure R14. The setup of Brownian dynamics (BD) simulations. (a) Schematic illustration of the CG models for *R*-pempH⁻ (*R*-pempH₂) molecule and Tb³⁺. Snapshots of CG model for chain-II (b) and chain-III (c) with six Tb³⁺ ions and eighteen *R*-pempH⁻ (*R*-pempH₂) molecules. To better differentiate the chain-II and chain-III, the bead that can interact with Tb³⁺ is used as pink and lime one, respectively. (d) The initial conformation of the self-assembly system, where the chains-II (and/or chains-III) are distributed uniformly in the simulation box.

To better clarify the experimental observations, we further used Brownian dynamics simulation to investigate the system from the mesoscopic view (Figure R14, Figure 8 in the new version). In the case of pure chain-III, rod-like chains with one or several monomers in the diameter were observed (Figure R15a, Figure 9a in the new version). While in the case of pure chain-II, due to the strong side interaction, block-like aggregates with similar length and height were observed (Figure R15b). These results were in accordance with the inference by all-atom simulation, and consisted with the experimental findings. More importantly, in the case of the mixture of chain-III and chain-II (the ratio is about 4:1), neither the rod-like chains nor block-like aggregates occurred. Instead, the curved or twisted chains were found (Figure R15c). Since the molecular symmetry of chain-III and chain-II was different (chain-III is *P6*₅ and chain-II is *P2*₁), when chain-II binded to chain-III along the axis,

the assembled chain would become a bit curved instead of linear growth in **R-3** system. Moreover, since the side interaction between chain-II and chain-II was much larger than that between chain-II and chain-II/chain-III, chain-II preferred to aggregate with each other, which further made the growth along the chain direction more curved and twisted. As demonstrated theoretically by Grason et al. [*Nat. Mater.*, **15**, 727-732 (2016)], when there existed many self-twisting chains or bundles, the chiral filaments may occur due to the frustration of inter-filament spacing. As a result, the macroscopic scales of chirality could be observed in this coordination polymer system.

Figure R15. Typical snapshots for the assembly in different systems. (a) Pure chain-III system; (b) pure chain-II system; (c) the chain-III and chain-II are mixed as the ratio of 4:1 in the presence of NO_3^- ; (d) the chain-III and chain-II are mixed as the ratio of 4:1 in the absence of NO_3^- ; (e) the chain-III and chain-II are mixed as the ratio of 1:1 in the presence of NO_3^- ; (f) the chain-III and chain-II are mixed as the ratio of 16:1 in the presence of NO_3^- . The ochre beads in the $R\text{-pempH}^+ / R\text{-pempH}_2$ molecules are not shown for clarity.

Additionally, if there did not exist the NO_3^- ions in the system, namely, the side interaction between chain-II and chain-II was the same as that between chain-III and chain-II/chain-III, the aggregation of chain-II in the assembly would not become obvious, and finally the assembly was nearly linear chain instead of twisted one (Figure R15d), which indicated the importance of NO_3^- in the chiral assembly and was in good agreement with the experimental findings.

Besides, we also found that if the mixture ratio of chain-III and chain-II was lower (e.g., 1:1), since the side interaction between chain-II and chain-II was large, block-like aggregates were again observed (Figure R15e). On the contrary, if the mixture ratio of chain-III and chain-II was much greater (16:1), since there existed no enough chains-II to induce the curved growth of chains-III, the twist or curve of the assembly was not obvious (Figure R15f), indicating that the ratio of chain-III and chain-II was also of great importance in helices formation. Considering that pH is related to the ratio of chain-III and chain-II in the system, our simulation results here could also clarify the role of pH in the experiments.

Figure R16. Proposed formation mechanism of crystalline materials of **R-3** (top) and **R-2** (middle), and helices of **R-1** (bottom).

On the basis of the experiment and simulation results, we believe that the present proposed mechanism is clear and we have also redrawn the schematical illustration of formation mechanism of crystalline material **R-3**, **R-2** and helices of **R-1** in Figure R16 (Figure 10 in the new version of the manuscript).

(2) Since the proposed mechanism is important for this manuscript, the introduction should not only list previous examples of chiral macrostructures, but also examine the proposed

mechanisms by which these morphologies are formed from chiral molecular components. Comparing refs 21 & 22, the mechanisms suggested seem to be quite varied and ill-defined.

Reply: We appreciate the reviewer for the suggestion. Accordingly, we have revised the introduction part by including the proposed mechanisms in refs 21 and 22 by which these morphologies are formed from chiral molecular components.

The description in the new version is: "By utilizing chiral amino acids, Tang et al. obtained homochiral Ag(I)/cysteine helical nanobelts in which the chirality transcription occurs from cysteine molecule to the assembly entities. The proposed mechanism involved the merging of nonhelical nanobelts of Ag(I)/cysteine layers through lateral attachment, which developed into the hierarchical helices with a specific twist direction. Huang et al. isolated right-handed Ca-cholate helical nanoribbons, which were further used as templates for the fabrication of helical inorganic nanomaterials. The formation mechanism was supposed to involve the twist of supramolecular layers composed of cholate bilayer strips connected via calcium-carboxyl coordination."

In both ref. 21 and 22, the coordination polymers were supposed to have layer structures, although single crystal data of these materials were not obtained. The assembly of the layers led to the formation of nanobelts, which further developed into nanohelices due to the twist of the nanobelts. In contrast, the terbium coordination polymers in the present work possess chain structures, demonstrated by the single crystal structural determinations. As did in ref. 21 and 22, with the aid of both experiments and molecular simulations, the formation mechanism is also proposed in the revised manuscript, namely, the assembly of two or three kinds of chains (neutral and positively charged) triggered the twist of chain growth. This mechanism is obviously different from that proposed in the previous works.

(3) Much of the detail given in the manuscript is peripheral to what I regard as the core point of importance – that of the generation of macroscale chirality. The introduction is devoted largely to a description of the general class of materials and the results and discussion are devoted largely to description of the crystal structures. Overall, the feeling as a reader is that there is something very interesting going on here, but then disappointment that the work hasn't progressed to a level where a more detailed and definitive mechanism can be determined and then contrasted with other instances of this type of effect.

Reply: Thanks for the comment. After a major revision, we hope that the current version of the manuscript has given enough details and clarified the questions.

(4) Figure 6 should make clear which of the chains shown are Chains-I, -II or -III.

Reply: We have added "Chains-I, -II, -III" in Figure 6 (Figure 10 in the new version of the manuscript). Thanks.

For Reviewer #2:

Understanding the transfer or amplification of chiral information within different scales, such as from molecular to macroscopic level, still remains a topic of great interest and activity. For coordination polymers, fabrication of chiral assemblies with helical structures taking the form of both single crystal and microscale architectures is difficult. This paper demonstrates the chirality transfer of artificial coordination polymers with very nice single crystal data, which might have chance to be published in nature communication. However, the following issues must be addressed before the final determination for acceptance.

Reply: We appreciate the reviewer for the positive and encouraging comments.

(1) The formation of single crystals with helical morphologies, especially the impact from pH values and anions, has been clearly discussed in the manuscript. The mechanism concerning chirality transfer from molecular assembly to micrometer scale, however, seems to be still not clear and lack of enough experimental evidence. For example, the manuscript shows the SEM images of the microscale helices with the diameters of about 6-9 μm and the helical crystal structures with the diameters less than 2 nm. However, the morphologies of the assemblies around nanoscales were not investigated. As a result, there appears no enough obvious evidence for supporting the formation of hierarchical bundles from molecular scales to nanoscales. I suggest the authors to perform the TEM measurement for these systems, which might provide more clear and useful information in this regard.

Reply: We appreciate the reviewer for raising these important questions. In order to clarify the formation mechanism, we have repeated the experiments and collected intermediate products before (0-min) and after hydrothermal reactions for 5 min, 10 min, 15 min, 20 min, 30 min, 1 h, 2 h, 4 h, 6 h, 8 h, 10 h, 20 h and 48 h. These products were characterized by PXRD, IR and SEM measurements, based on which the formation mechanism was proposed (please see details in the response to Q1 of Reviewer #1). In addition, molecular simulations were also performed to support the proposed mechanism.

Indeed, the mechanism concerns chirality transfer from molecular assembly to nano- and then micro-meter scale. The diameters of single chains in compounds **R-2** and **R-3** are all less than 2 nm. While the microscale helices were found in the reaction products, the widths of which reached about 6-9 μm after hydrothermal reactions for 2 days. In order to observe

the nanoscale assemblies, we isolated the solid products with much shorter hydrothermal reaction time, e.g. 5 min, 10 min, 15 min, 20 min, 30 min and 1 h. Although the PXRD measurements suggested the formation of **R-1** after hydrothermal reactions for 10 min, the SEM images showed only nano-particles without helical morphologies. Small helices of **R-1** were observed in the 20-min product with the widths and lengths of ca. 150-500 nm and 2-5 μm , respectively (Figure R7, top). With increasing reaction time, the widths of the helices increase to 190-970 nm for 30-min (Figure R8), and 0.19-2.26 μm for 1-h (Figure R9). Obviously, the helical morphologies of the assemblies were developed from nano- to micro-meter scale.

We also tried to measure the TEM spectra of the helices, as shown in Figure R17. It seems that in the present cases, the TEM images cannot give more information than SEM images.

Figure R17. The TEM image of the 1-h product.

(2) For the solid-state samples, circular dichroism (CD) spectral measurement should be cautious. The LD artifacts should be considered. Please provide details of the experimental conditions of corresponding spectral measurements.

Reply: We appreciate the reviewer for raising this important question, which was often ignored by most authors. The simultaneous CD and LD measurements have been conducted for **R-1** and **S-1** using the multi-probe function in the J-1500 CD spectrometer. Each sample was diluted by KCl with a ratio of 1/50 and pressed into a pellet. **R-1** or **S-1**, the angle dependence behavior of all eight CD spectra, which were measured in a step of 45° about the optical axis, has been checked (Figure R18, Supplementary Figure 7 in the new version)

and the average CD spectra were shown in Figure R19 (Supplementary Figure 8 in the new version). The Cotton effects centred at 262 nm and opposing symmetries were observed for **R-1** and **S-1**. The result was consistent with the previous CD measurements. We also compared the intensity of CD and LD signals in the same unit of [dOD] (Figure R20, Supplementary Figure 9 in the new version). The peak height of CD signals was ca. 3 times that of LD signals, confirming that the observed CD spectra were true.

Figure R18. The angle dependence CD spectra of **R-1** and **S-1**.

Figure R19. The average CD, LD, HT and absorption spectra of **R-1** and **S-1**.

Figure R20. The comparison of CD and LD spectra in the same [dOD] unit for *R-1* and *S-1*.

(3) In addition, more detailed experimental conditions should be provided. For example, the pH value of the systems should be measured both before and after hydrothermal reactions.

Reply: Thanks for raising the good question. We have performed 16 batches of parallel reactions. A suspension of $\text{Tb}(\text{NO}_3)_3$ and *R*-pempH₂ was stirred at room temperature for 2 hours and then adjusted to pH 3.0 – 3.3 using 0.5 M NaOH. After heating at 120 °C under hydrothermal conditions for 2 days, the white (flocculent) precipitates were obtained. The pH values of the resulted solutions fall in a range of 2.5 to 2.8 (Figure R21, Supplementary Figure 41 in the new version). Apparently, the pH dropped to a lower value after hydrothermal reactions of $\text{Tb}(\text{NO}_3)_3$ and *R*-pempH₂.

Figure R21. The pH value of the systems measured before and after hydrothermal reactions of $\text{Tb}(\text{NO}_3)_3$ and *R*-pempH₂ for 2 d.

For Reviewer #3:

This is an interesting manuscript reporting on the synthesis and characterization of a chiral terbium-based coordination polymer. The work is sound and well reported. I think it is suitable for publication in Nature Communications after some modifications, to improve the understanding of the system. Right now, this is rather a technical report, with all the data gathered but not much in terms of physical and chemical understanding of the system. I would suggest three improvements to the work before publication:

Reply: We appreciate the reviewer for considering the work as "an interesting manuscript" and "suitable for publication in Nature Communications after some modifications".

(1) The visualization of the coordination (e.g. Figure 3a) is rather poor, and I have a very hard time understanding the nature of the coordination pattern around the metal cations.

Reply: Thanks for the question. We have re-drawn the figure, from which the coordination around the metal cations should be better visualized.

(2) The formation of "self-twisting bundles" (Figure 6 and related text) is not convincingly backed by data. Right not, it sounds more like a hypothesis of the authors. I suggest they gather data to prove this hypothesis. Especially as it features heavily in the manuscript, and in the graphical abstract too.

Reply: We appreciate the reviewer for raising this important question. In order to clarify the formation mechanism, we have repeated the experiments and collected intermediate products before (0-min) and after hydrothermal reactions for 5 min, 10 min, 15 min, 20 min, 30 min, 1 h, 2 h, 4 h, 6 h, 8 h, 10 h, 20 h and 48 h. These products were characterized by PXRD, IR and SEM measurements, based on which the formation mechanism was proposed. In addition, molecular simulations have also been conducted to support the mechanism (please see details in the response to Q1 of Reviewer #1).

(3) More chemical and physical analysis of the system needs to be given. Right now, the study convincingly shows the formation of this chiral CP, but does not explain to the reader why it is rare and specific to this particular system. Maybe an analysis of the packing or contact surfaces, or computations, could give a better physical understanding of the system?

Reply: We appreciate the reviewer for raising the questions and giving the suggestions. In the new version, we have provided more experimental details including the PXRD, IR, SEM and EDX analyses of the intermediate states. Molecular simulations have also been

performed to support the proposed mechanism (please see details in the response to Q1 of Reviewer #1).

The most significant feature of this work is two fold: (1) it provides a rare case of coordination polymers that can be assembled in both crystalline and helical forms. As far as we are aware, similar work has never been reported before. (2) The mechanism revealed in this work is unique, and has never been described before. That is, the assembly of two or three kinds of chains (neutral and positively charged) triggered the twist of chain growth, and the nitrate anion plays an important role in promoting the assembly of different types of chains.

Reviewers' comments:

Reviewer #1 (Remarks to the Author):

In responding both to my previous comments, and those of the other two reviewers, the authors have performed a large body of additional syntheses, experimental characterisations and theoretical simulations, and rewritten large parts of the manuscript so as to better describe relevant previous work and to better focus on the key point of novelty – that of the transference of chirality from the molecular scale to the macroscale. In assessing the revised manuscript I have been keen to ensure that the observed phenomenon has been described and delineated clearly, and that a suitably advanced understanding has been arrived at, the latter being quite challenging given the structural complexity of the macroscale helices.

The new syntheses are a welcome addition to the manuscript in providing some better understanding of the various conditions that favour helix formation, in turn serving to shed some light on the microscopic mechanism. IR analysis, which has led to the identification of a suspected role of uncoordinated nitrate ions in the formation of intertwining chains, has also been a very useful addition. Accompanying this, microscale EDX analysis has usefully shown that compositional heterogeneity exists within the helices, and simulations have been performed to explore the interaction strengths between the various components.

As is perhaps to be expected given the macroscale complexity of these helices, despite these new characterisations and simulations there remains considerable uncertainty as to the nature of the underlying mechanism. While I find it a little unusual to read such a high level of speculation in a manuscript pitched at this level of publication, I'm happy to regard this as appropriate in the circumstances. I would reiterate that there remains great scope for microscale diffraction-based analyses to be performed on the helices, and that this would likely lead to considerable further insight into the mechanism of helix formation; however, given that this is a rather specialised technique, and given that the authors have performed a range of other characterisations and simulations to provide clues on this point, I am happy to regard such diffraction investigations as being appropriate for future work rather than an essential for publication of the current work.

In light of the above, and the fact that this system has a number of unique aspects over those reported in references 21 and 22, I am happy to recommend publication in Nature Communications. The manuscript would benefit from some further grammatical and scientific tightening, which might hopefully be possible at the editorial stage.

Reviewer #2 (Remarks to the Author):

I am happy to see that the authors have provided detailed and apparently satisfactory responses to the extensive comments of the referees. Especially, in order to clarify the formation mechanism, they performed the hydrothermal reactions with different length of time, and collected 14 different types of intermediate products. The detailed characterization of these products can support the proposed mechanism. Moreover, the simulation part in the revised manuscript is also fine. My own opinion is that the paper should now be accepted for publication in Nature Communications.

Reviewer #3 (Remarks to the Author):

In the revised manuscript, the authors have added a full section on molecular simulations to provide microscopic insight into the mechanism by which macroscopic chirality can occur. There are limitations to the methodology used, which need to be addressed before the manuscript can be

considered for publication.

1. The authors used all-atom MD simulations to provide parameters for their coarse-grained Brownian dynamics. However, from the all-atom simulations they only extract average ion/ligand interaction energies. In order to show that the coarse-grained model does represent well the coordination characteristics of the system, they need to provide a comparison of the structure and dynamics of the all-atoms and coarse-grained descriptions. In particular, radial and angular distribution functions around the Tb^{3+} ions need to be compared and validated.
2. Although the authors see some larger-scale structure arising from the Brownian dynamics simulations, it is clearly still far from macroscopic scale. This is thus not direct evidence of the macroscopic mechanism. This needs to be explicitly stated and the length scales of the microstructures observed in BD simulations need to be given (in text and in Figure 9).
3. Finally, the authors note that the coarse grained simulations give rise to "curved or twisted chains". But are these structures actually chiral? On the only figure presented (Figure 9, and in particular panel c) it looks more like a rod or chain with some random fluctuations of shape. This is far from being the same as a chiral helix.

Response to the referees' questions:

For Reviewer #1:

In responding both to my previous comments, and those of the other two reviewers, the authors have performed a large body of additional syntheses, experimental characterisations and theoretical simulations, and rewritten large parts of the manuscript so as to better describe relevant previous work and to better focus on the key point of novelty – that of the transference of chirality from the molecular scale to the macroscale. In assessing the revised manuscript I have been keen to ensure that the observed phenomenon has been described and delineated clearly, and that a suitably advanced understanding has been arrived at, the latter being quite challenging given the structural complexity of the macroscale helices.

The new syntheses are a welcome addition to the manuscript in providing some better understanding of the various conditions that favour helix formation, in turn serving to shed some light on the microscopic mechanism. IR analysis, which has led to the identification of a suspected role of uncoordinated nitrate ions in the formation of intertwining chains, has also been a very useful addition. Accompanying this, microscale EDX analysis has usefully shown that compositional heterogeneity exists within the helices, and simulations have been performed to explore the interaction strengths between the various components.

As is perhaps to be expected given the macroscale complexity of these helices, despite these new characterisations and simulations there remains considerable uncertainty as to the nature of the underlying mechanism. While I find it a little unusual to read such a high level of speculation in a manuscript pitched at this level of publication, I'm happy to regard this as appropriate in the circumstances. I would reiterate that there remains great scope for microscale diffraction-based analyses to be performed on the helices, and that this would likely lead to considerable further insight into the mechanism of helix formation; however, given that this is a rather specialised technique, and given that the authors have performed a range of other characterisations and simulations to provide clues on this point, I am happy to regard such diffraction investigations as being appropriate for future work rather than an essential for publication of the current work.

In light of the above, and the fact that this system has a number of unique aspects over those reported in references 21 and 22, I am happy to recommend publication in Nature Communications. The manuscript would benefit from some further grammatical and scientific tightening, which might hopefully be possible at the editorial stage.

Reply: We appreciate the reviewer for the valuable and positive comments, and for the recommendation of publication of this work in Nature Communications.

For Reviewer #2:

I am happy to see that the authors have provided detailed and apparently satisfactory responses to the extensive comments of the referees. Especially, in order to clarify the formation mechanism, they performed the hydrothermal reactions with different length of time, and collected 14 different types of intermediate products. The detailed characterization of these products can support the proposed mechanism. Moreover, the simulation part in the revised manuscript is also fine. My own opinion is that the paper should now be accepted for publication in Nature Communications.

Reply: We appreciate the reviewer for the positive comments, and for the recommendation of publication of this work in Nature Communications.

For Reviewer #3:

In the revised manuscript, the authors have added a full section on molecular simulations to provide microscopic insight into the mechanism by which macroscopic chirality can occur. There are limitations to the methodology used, which need to be addressed before the manuscript can be considered for publication.

1. The authors used all-atom MD simulations to provide parameters for their coarse-grained Brownian dynamics. However, from the all-atom simulations they only extract average ion/ligand interaction energies. In order to show that the coarse-grained model does represent well the coordination characteristics of the system, they need to provide a comparison of the structure and dynamics of the all-atoms and coarse-grained descriptions. In particular, radial and angular distribution functions around the Tb^{3+} ions need to be compared and validated.

Reply: We appreciate the reviewer for raising this important question, and we are also very sorry for missing some modeling details in BD simulation.

In BD simulation, the initial coordinate of Tb beads and P1 beads (in the monomer) was obtained by using the coordinate of Tb^{3+} ion and P atom (of *R*-pempH⁻ molecule) in experimental crystal data, respectively. Moreover, Tb beads and P1 beads in each monomer were treated as rigid ones in BD simulation, namely, the relative position of Tb beads and P1 beads in the monomer was kept fixed, which was due to the following two reasons. Firstly, as shown in Fig. R1, the all-atom MD simulation results showed that the relative position of Tb^{3+}

ion and P atom changed very little since the peak of the Tb-P distance and the P-Tb-P angle was very sharp in the probability distribution functions. Actually, the values of these parameters in all-atom MD simulations were close to those in the experiments (see the caption in Fig. R1). Secondly (and honestly speaking), if we did not constrain the relative position of Tb beads and P1 beads in the monomer, the molecular structure obtained in BD simulation could be totally different from that in all-atom MD simulation, and more importantly, differed from the unit in the experimental structure (e.g., chain II and chain III cannot be distinguished). As a result, here we treat the Tb beads and P1 beads in each monomer as rigid ones in the BD simulations.

In the new version of the manuscript, we have added the above description of modeling details on pages 26-27.

Fig. R1 The cumulative probability distribution of Tb-P distance (a) and P-Tb-P angle (b) in Tb^{3+} -*R*-pempH⁻ system in all-atom simulation (averaged over 100 ns). The peaks in (a) are 2.99 Å, 3.23 Å, 3.39 Å, respectively; while the Tb-P distances in the experiments are 3.07 Å, 3.13 Å, 3.55 Å, respectively. The peaks in (b) are 85.50°, 96.50°, 174.50°, respectively; while the P-Tb-P angles in the experiments are 83.00°, 98.66°, 173.35°, respectively.

2. Although the authors see some larger-scale structure arising from the Brownian dynamics simulations, it is clearly still far from macroscopic scale. This is thus not direct evidence of the macroscopic mechanism. This needs to be explicitly stated and the length scales of the microstructures observed in BD simulations need to be given (in text and in Figure 9).

Reply: This is a good comment! Due to the limitation of present computing technology, now we still cannot perform the simulations at the macroscopic scale. In the BD simulations, the length scale of the obtained chains was about tens of (at most one hundred) nanometers, which was indeed far from the length scale of the helices (tens of micrometers) in the experiments. In this sense, and as pointed out by you, the simulation results here should not

be the direct evidence of the macroscopic mechanism. However, the main purpose of the MD and BD simulations here is to give some hints for the mechanism of helix formation in experiments. Actually, by combining MD and BD simulations, we have given some possible explanations for the formation of helices and block- or rod-like crystalline materials under different situations; in addition, the effect of pH and NO_3^- ion on the self-assembly in the simulations consists well with that in experiments. In general, although the direct evidence is not given here, the simulations provide some clues on the underlying mechanism of helix formation (as mentioned by Reviewer 1).

According to your suggestion, we have explicitly stated the above issues on page 17, and also given the length scales of the microstructures observed in BD simulations in Figure 9 in the new revision of the manuscript.

3. Finally, the authors note that the coarse grained simulations give rise to "curved or twisted chains". But are these structures actually chiral? On the only figure presented (Figure 9, and in particular panel c) it looks more like a rod or chain with some random fluctuations of shape. This is far from being the same as a chiral helix.

Reply: This is also a very good comment! To quantitatively depict the chirality of the obtained structure, we calculate the tangent correlation function of the chain along the z direction [J. Chem. Phys., 134, 065107 (2011)]: $\eta = \mathbf{t}_z \cdot \mathbf{t}_0$, where \mathbf{t}_z is the mean tangent vector at the coordinate z.

For ideal helical chain (i.e., helical chain at zero temperature), the tangent correlation function has the form of cosine function: $A + B\cos(kz)$. Previous theory also indicated that, at finite temperature, (due to thermal fluctuation) the correlation function of the helical chain has the form: $Ce^{-k_1z} + De^{-k_2z}\cos(k_3z)$ [J. Chem. Phys., 134, 065107 (2011)]. Though the obtained structure seems to be far from the ideal chiral structure, it may have some chirality since there exists some similarity of the correlation function profile between the structure here and the helical chain at finite temperature (see Fig. R2). Moreover, as we know, the curvature of the chain should be the prerequisite for the helix formation; in other words, the linear chain can never form the helix. Actually, as shown in Fig. R2, the correlation function of the linear chain (kept as 1.0 with the increase of z) is totally different from the curved chain in (b) and the helical chain in (a). In general, the curved chain in Figure 9c is different from the linear chain in Figure 9a, and may have some chiral property, thus it may be used to explain the possible mechanism of helix formation in the experiments (although it is not the direct evidence).

In the new version of the manuscript, we have given some additional discussions on this on page 17.

Fig. R2 Tangent correlation functions (η) for different types of chains. (a) Theoretical results of tangent correlation functions for helical chain at finite temperature. (b) Simulation results of tangent correlation functions for linear chain (red line) in left panel of Figure 9a and curved chain (black line) in right panel of Figure 9c.

REVIEWERS' COMMENTS:

Reviewer #3 (Remarks to the Author):

I am satisfied with the authors' modifications to the manuscript, which I believe is now suitable for publication as is.